# *World-Shaper*: A Unified Framework for 360° Panoramic Editing

**Dong Liang** [* 1 2]  **Yuhao Liu** [* 1]  **Jinyuan Jia** [2]  **Youjun Zhao** [1]  **Rynson W.H. Lau** [1 3]

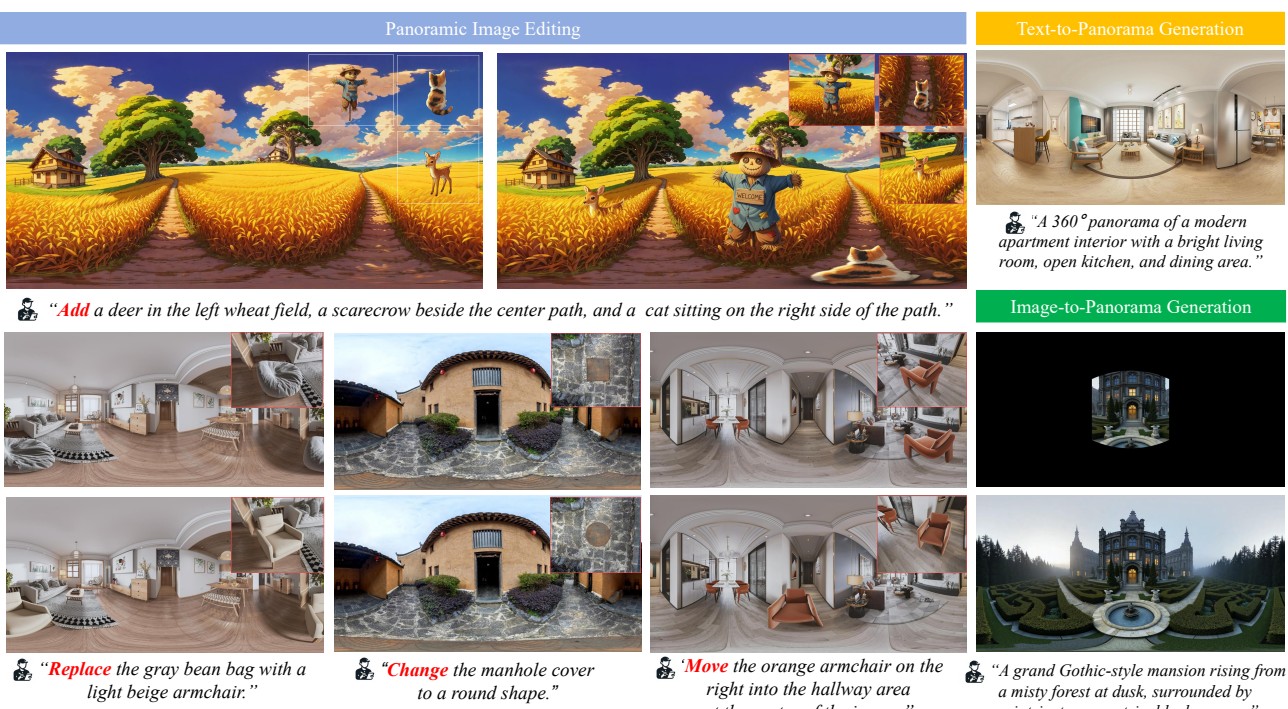

Figure 1. Our geometry-aware diffusion framework, **World-Shaper**, unifies panorama generation and editing within a single editing-centric model, supporting text-to-panorama and image-to-panorama generation, and diverse editing operations such as object addition, removal, replacement, relocation, and appearance modification. New objects and edited regions are highlighted in white and red boxes, respectively.

## Abstract

Being able to edit panoramic images is crucial for creating realistic 360° visual experiences. However, existing perspective-based image editing methods fail to model the spatial structure of panoramas. Conventional cube-map decompositions attempt to overcome this problem but inevitably break global consistency due to their mismatch with spherical geometry. Motivated by this insight, we reformulate panoramic editing directly in the equirectangular projection (ERP) domain and present **World-Shaper**, a unified geometry-aware framework that supports five distinct editing operations within a single ERP-native representation. To address the latitude-dependent geometric distortion inherent in ERP, we introduce a geometry-aware learning strategy comprising distortion-aware attention modulation (DAAM), which steers cross-attention with latitude-dependent strength at the feature level; layered shape loss (LSL), which enforces per-object geometric supervision at the output level; and progressive curriculum training to internalize panoramic priors. To overcome the scarcity of paired panoramic editing data, we train a dedicated ERP-native controllable generator that synthesizes objects directly in the equirectangular domain under user-defined conditions, enabling

[1]City University of Hong Kong, Hong Kong SAR, China. [2]Tongji University, Shanghai, China. [3]City University of Hong Kong, Dongguan, China. Correspondence to: Yuhao Liu <yuhaoliu7456@gmail.com>, Jinyuan Jia <jyjia@tongji.edu.cn>, Rynson W.H. Lau <Rynson-Lau@gmail.com>.

*Proceedings of the $43^{rd}$ International Conference on Machine Learning*, Seoul, South Korea. PMLR 306, 2026. Copyright 2026 by the author(s).

scalable paired data construction for learning diverse editing behaviors. Extensive experiments on our new benchmark, ***PEBench***, demonstrate that ***World-Shaper*** achieves superior geometric consistency, editing fidelity, and text controllability compared to state-of-the-art methods, enabling coherent and flexible 360° visual world creation with unified editing control. Code, models, and data are available at the project page.

## 1. Introduction

Unlike a perspective image, which depicts only a limited field of view from one direction, a panorama captures the *entire visual world* surrounding the observer, offering a complete 360° perception of the scene. This makes it indispensable for applications such as immersive media (Tukur et al., 2025; Lin et al., 2025a), environmental lighting (Wang et al., 2022; Cheng & Ji, 2024), VR/AR (Wang et al., 2025a; Bai et al., 2023), and autonomous driving (Chugunov et al., 2024; Lin et al., 2025a). Being able to generate and edit such panoramic worlds is therefore essential for creating and controlling realistic 360° experiences. However, editing in this 360° domain remains challenging.

Despite the rapid progress in diffusion-based image generation (Rombach et al., 2022; Labs, 2024; Peebles & Xie, 2023; Achiam et al., 2023) and editing (Brooks et al., 2023; Batifol et al., 2025a; Wang et al., 2025c; Zhang et al., 2025c), these models are fundamentally designed to handle perspective images. Panoramic images, on the contrary, are commonly represented in the equirectangular projection (ERP) (Lin et al., 2025b), which maps the spherical environment to a 2D plane using latitude–longitude coordinates. When applied directly to the ERP-based panoramas, perspective-based models struggle to deal with the spatial distortion, as shown in Fig. 2(b). A trivial workaround is to convert the panorama into multiple cube-map faces, process each face independently, and then stitch them back together. Such a decomposition, however, inevitably breaks global coherence: a single object may span multiple faces, as shown in Fig. 2(c) causing geometry and texture discontinuities when the cube faces are reassembled, as shown in Fig. 2(d).

Based on the above observation, we explore in this work an ERP-native formulation for panoramic editing, which offers two key advantages: (1) ***Global Consistency*** – the model perceives the entire 360° environment at once, enforcing semantic and geometric continuity across all directions and eliminating cross-face seams; and (2) ***Unified Representation*** – a single, continuous coordinate system aligns naturally with pre-trained diffusion models (Wu et al., 2025a), forming an editing-centric representation where text prompts and spatial conditions apply consistently. However,

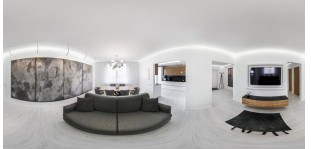
(a) Input Panoramic Image

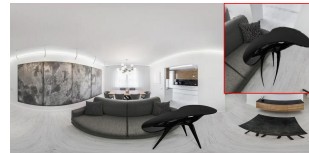
(b) Nano Banana Pro (Google, 2025)

*"Add a black table in front of the sofa."*

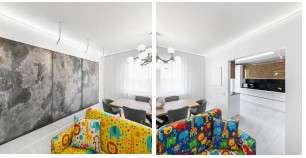
(c) Cube-maps after editing

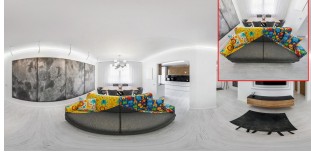
(d) Reprojection back to ERP

*"Change the sofa's pattern to a children's theme."*

*Figure 2.* Two paradigms for editing panoramic images using existing perspective-based models. First row: applying Nano Banana Pro (Google, 2025) directly to ERP panoramas suffers from severe spatial distortion. Second row: locally editing cube-map faces and reprojecting them back to ERP reduces distortion but breaks global coherence, causing discontinuities across cube boundaries. Refer to Sec. D.2 of Supplemental for more discussions.

realizing this goal is far from trivial, due to the uneven geometric distortion in ERP images and the lack of paired data for panoramic editing.

To address these two challenges, we propose ***World-Shaper***, a unified geometry-aware framework for direct panoramic image editing. To handle geometric distortions, we introduce a *geometry-aware learning strategy* that comprises three components: (i) ***distortion-aware attention modulation*** (DAAM), which steers cross-attention with latitude-dependent strength at the feature level; (ii) ***layered shape loss*** (LSL), which enforces per-object geometric supervision at the output level; and (iii) ***progressive curriculum training***, which gradually shifts training from global panorama generation to localized object-level manipulation, allowing the model to internalize panoramic distortion priors. To address the lack of paired panoramic editing data, we further train an ERP-native controllable generator to synthesize objects directly in the equirectangular domain under user-defined conditions, enabling scalable paired data construction for diverse editing behaviors. Through this unified design, ***World-Shaper*** enables panorama generation and diverse editing capabilities such as object addition, removal, relocation, replacement, and appearance modification.

To verify the effectiveness of our ***World-Shaper***, we curate a new panoramic editing benchmark, ***PEBench***, and conduct extensive experiments across object-level panoramic editing and generation. ***World-Shaper*** demonstrates superior geometric consistency, editing fidelity, and text controllability on both synthetic and real-world panoramas. Beyond quantitative results, our framework empowers intuitive and flexible 360° visual world creation and editing. Furthermore, our framework can be progressively extended toward 3D world generation, enabling scene expansion and exploration

from user-provided text or images.

Our main contributions can be summarized as follows:

- We introduce **World-Shaper**, a unified framework that supports five distinct panoramic editing operations within a single ERP-native representation, ensuring global consistency and seamless cross-view editing.
- We propose a geometry-aware learning strategy to handle latitude-dependent distortion in ERP, comprising (i) distortion-aware attention modulation (DAAM), which steers cross-attention with latitude-dependent strength at the feature level; (ii) layered shape loss (LSL), which enforces per-object geometric supervision at the output level; and (iii) progressive curriculum training to internalize panoramic priors.
- We curate **PEBench**, a comprehensive panoramic editing benchmark, and conduct extensive experiments to demonstrate the superiority of our method.

## 2. Related Works

### 2.1. Text-to-Image Generation

Recent advances in diffusion model (Ho et al., 2020; Song et al., 2021) have fundamentally reshaped text-to-image (T2I) generation. Early diffusion-based frameworks such as DALL·E (Ramesh et al., 2021; 2022; Betker et al., 2023), Imagen (Saharia et al., 2022), and Stable Diffusion (Rombach et al., 2022; Podell et al., 2023) established the foundation for scalable and high-quality image synthesis conditioned on language descriptions. More recently, DiT-based architectures have become the dominant paradigm. Representative models including SD3 (Stability AI et al., 2024), FLUX.1 (Labs, 2024), Qwen-Image (Wu et al., 2025a), and Seedream (Seedream et al., 2025) have demonstrated strong generative quality across diverse domains. Yet text-only conditioning provides limited spatial and structural control, which has led to an increasing interest in controllable T2I frameworks (Zhang et al., 2023; Li et al., 2024a; Mou et al., 2023; Zhao et al., 2023; Zhang et al., 2025a; Tan et al., 2025; Liang et al., 2025) to leverage additional modalities. Building on this, we adopt a controllable panoramic generation as an auxiliary stage of our generate-then-edit pipeline, enabling the construction of paired panoramic data for editing model training.

### 2.2. Instruction-based Image Editing

Image editing aims to find an optimal balance between image reconstruction and re-generation. Early approaches (Hertz et al., 2023; Chefer et al., 2023; Brooks et al., 2023; Abdal et al., 2019; Roich et al., 2022; Tumanyan et al., 2023) primarily operate in a training-free manner, relying on attention manipulation (Hertz et al., 2023; Chefer et al., 2023), prompt engineering (Brooks et al., 2023), or

latent-space inversion (Mokady et al., 2023; Cao et al., 2023) to steer T2I models toward the desired edit. To overcome the weak robustness, subsequent works (Xiao et al., 2025; Wu et al., 2025c; Zhang et al., 2025c; Batifol et al., 2025b; Wu et al., 2025b; Sheynin et al., 2023; Liu et al., 2024b; Liang et al., 2026) adopt a data-driven paradigm: they first construct large-scale editing datasets (Zhao et al., 2024; Yu et al., 2025; Ye et al., 2025), and then train unified models to jointly understand instructions and generate edited images. More recently, methods like (Wu et al., 2025b; Liu et al., 2025; Achiam et al., 2023; Google, 2025) incorporate multimodal understanding into diffusion backbones, equipping editing models with world-level reasoning and yielding more consistent, semantically aligned edits. Despite their success, these methods are all based on perspective images and fail to handle spatial distortions inherent in panoramic projections.

### 2.3. Panorama Generation and Editing

*Text-to-panorama* generation aims to synthesize panoramic images from textual descriptions. Existing methods can be broadly grouped according to how they address the geometric challenges of the ERP representation. Distortion-aware methods (Chen et al., 2022; Sun et al., 2025; Wu et al., 2024; Quattrini et al., 2024) modify networks or feature sampling to compensate for the non-uniform stretching. Projection-driven methods (Çapuk et al., 2025; Zhang et al., 2024; Park et al., 2026; Kalischek et al., 2025) mitigate distortions by converting panoramas into alternative views (*e.g.*, cubemap) and learning to fuse multi-projection features. Continuity-oriented methods (Liu et al., 2024a; Feng et al., 2023; Tang et al., 2023; Feng et al., 2025) focus on maintaining horizontal cyclic consistency. Beyond text-based synthesis, *image-to-panorama* (Akimoto et al., 2022; Wu et al., 2023; Zheng et al., 2025; Zhang et al., 2025b; Huang et al., 2025a; Wang et al., 2023; Schwarz et al., 2025; Yuan et al., 2025) generation further incorporates NFOV images as explicit local constraints. Further, *3D* (Xie, 2025; Team et al., 2025; Yang et al., 2025c; Huang et al., 2025b; Li et al., 2025) and *video-based* (Li et al., 2024b; Xia et al., 2025; Wang et al., 2024; Dong et al., 2025; Yang et al., 2024) panorama generation methods extend the traditional panorama generation task toward dynamic or explorable 360° environments generation.

Panorama editing methods like SE360 (Zhong et al., 2025) and Omni2 (Yang et al., 2025a) rely heavily on cubemap projections. Their dataset generates by cubemap-inpainting, which restricts its edits to simple object-level operations. In contrast, our method operates directly in the ERP domain, preserving spherical continuity during both data synthesis and model training, enabling globally consistent geometry and richer edit types.

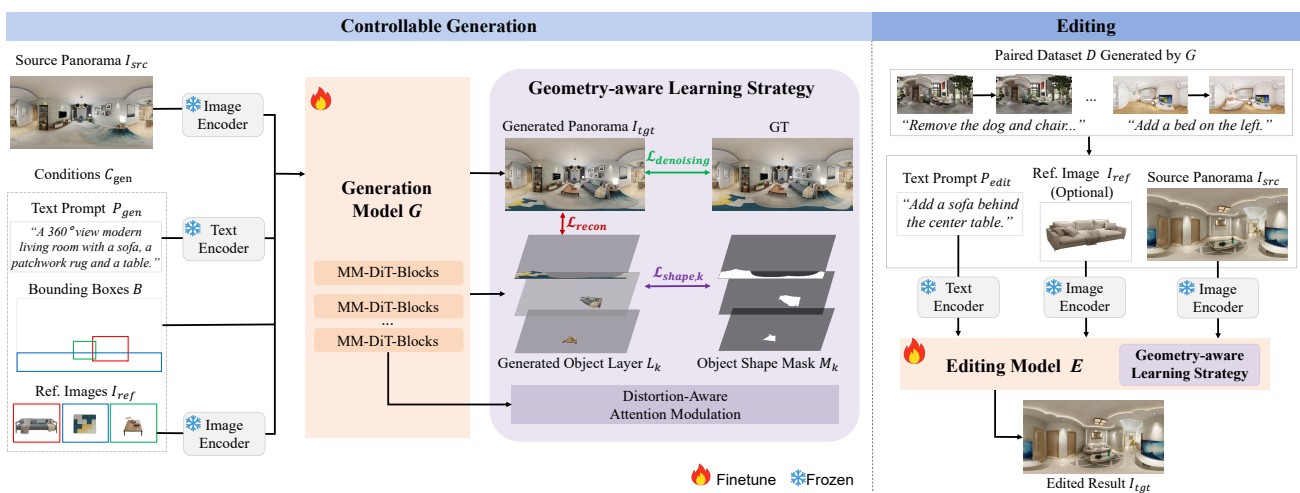

**Figure 3.** Pipeline of **World-Shaper**. It operates directly in the ERP domain and is centered on a geometry-aware learning strategy that handles latitude-dependent panoramic distortion through three components: distortion-aware attention modulation (DAAM), layered shape loss (LSL), and progressive curriculum training. To address the scarcity of paired panoramic editing data, we further train an ERP-native controllable generator $G$ to synthesize target panoramas $I_{tgt}$ from source panoramas $I_{src}$ under conditions $C_{gen}$. The synthesized pairs $(I_{src}, P_{edit}, I_{tgt})$ form a large-scale dataset $D$, on which the editing model $E$ is trained for instruction-driven panoramic editing.

## 3. Method

### 3.1. Overview

Our goal is to develop a unified panoramic image editing framework, denoted as **World-Shaper**, which integrates controllable panoramic generation and editing within a single model. This design enables coherent 360° panoramic image synthesis and diverse editing operations, such as object addition, removal, replacement, relocation, and appearance modification. Formally, given a source panorama $I_{src}$ in the equirectangular projection (ERP) format and an editing instruction $C_{gen}$ (*e.g.*, text prompt and/or spatial conditions), our model produces an edited panorama $I_{tgt}$ that satisfies the instruction while preserving global geometric consistency.

To achieve this, **World-Shaper** integrates two components. (1) *Geometry-Aware Learning Strategy*, which handles latitude-dependent ERP distortions via distortion-aware attention modulation, layered shape loss, and progressive curriculum training, enabling distortion-aware geometry learning from feature-level attention, output-level object supervision, and staged task learning. (2) *ERP-Native Data Synthesis*, where a controllable panoramic generation model serves as a data generator to synthesize constrained scenes and construct paired data for supervised editing learning. Fig. 3 shows the overall architecture.

### 3.2. Geometry-Aware Learning Strategy

To ensure geometrically consistent generation and editing, we introduce a geometry-aware learning strategy with three complementary designs: (1) *distortion-aware attention modulation*, which adapts object-specific cross-attention to latitude-dependent ERP distortion; (2) *layered shape loss*, which provides output-level per-object geometric su-

pervision via transparent object layers; and (3) *progressive curriculum training*, which shifts learning from global panorama generation to localized object manipulation to internalize panoramic distortion priors.

#### 3.2.1. DISTORTION-AWARE ATTENTION MODULATION

In ERP panoramas, object deformation varies with latitude. Therefore, at the feature level, we modulate object-specific cross-attention using the ground-truth object mask to guide object tokens toward the correct spatial support, with latitude-dependent strength that provides stronger guidance near highly distorted polar regions.

Specifically, we first extract the object-relevant cross-attention map $A$ between image tokens $\mathbf{X}$ and the object text embeddings $\mathbf{E}_o$. It is then modulated using the ground-truth object mask $M$ with spatially adaptive strength. We adopt a residual-based modulation (Kim et al., 2023), where the modulation residual is computed as:

$$R = \boldsymbol{\alpha} \odot (M \odot (A_{\max} - A) - (1 - M) \odot (A - A_{\min})), \tag{1}$$

where $\boldsymbol{\alpha}$ is a spatially adaptive modulation map determined by the latitude of each spatial location. For each pixel (or image token) at vertical coordinate $y$, we define:

$$\boldsymbol{\alpha}(y) = 1 - \cos\left(\frac{\pi}{2} - \frac{\pi y}{H}\right), \tag{2}$$

where $\boldsymbol{\alpha}(y)$ increases toward higher latitudes. In this way, the attention map is finally updated as $A' = A + R$.

#### 3.2.2. LAYERED SHAPE LOSS

At the output level, we introduce a layered shape loss to provide explicit per-object geometric supervision. Directly

supervising object geometry on the final blended panorama is unreliable, since the edited object is entangled with background texture and ERP distortion makes off-the-shelf mask extraction unstable. We therefore ask the model to additionally render each edited object as a separate transparent layer, so that its shape can be supervised independently before being composited back to the panorama.

Specifically, besides the generated panorama image, the model additionally outputs a set of $K$ transparent object layers $\{L_1, \ldots, L_K\}$, where each layer corresponds to one input object. Each layer $L_k$ is represented as an RGBA image, whose alpha channel provides a direct estimate of the object's shape mask $M_k$. In our implementation, since the employed VAE (Wu et al., 2025b) Decoder produces only RGB outputs, we approximate the RGBA representation by generating RGB images with a pure-white background and then derive the object's alpha mask $M_k$ via connectivity-based segmentation.

To supervise these layers, we impose two complementary constraints. First, we use an IOU loss $\mathcal{L}_{\text{shape},k} = \alpha_k \cdot (1 - \text{IoU}(M_k, M_{\text{shape},k}))$ to ensure that the predicted mask $M_k$ aligns with its GT shape mask $M_{\text{shape},k}$, where $\alpha_k$ is a spatially adaptive coefficient computed based on Eq. 2. Second, we introduce a reconstruction loss to enforce each predicted layer $L_k$ to be consistent with its appearance in the final panorama, by reconstructing the composite image $I_{\text{comp}} = \text{Composite}(I_{\text{src}}, L_1, \ldots, L_K)$ and matching it with $I_{\text{tgt}}$ within the object's bounding box $B_k$, as:

$$\mathcal{L}_{\text{recon}} = \frac{1}{N} \sum_{i=1}^{N} \|I_{\text{comp}}(i) - I_{\text{tgt}}(i)\|_2^2, \qquad (3)$$

where $N$ is the total number of pixels. The final layered shape loss aggregates both objectives across all $K$ objects:

$$\mathcal{L}_{\text{shape}} = \frac{1}{K} \sum_{k=1}^{K} (\mathcal{L}_{\text{shape},k}) + \mathcal{L}_{\text{recon}}. \qquad (4)$$

This layered supervision delivers explicit geometric guidance, prompting accurate and distortion-aware object shapes across the entire panorama.

### 3.2.3. PROGRESSIVE CURRICULUM TRAINING

While the distortion-aware attention modulation and the layered shape loss explicitly regulate geometric distortion, they remain limited by the diversity of training instances. In contrast, global panoramic generation tasks naturally expose the model to abundant and diverse distortion configurations. To leverage these rich priors, we gradually transition learning from global panorama generation to localized object manipulation, enabling the model to internalize panoramic distortion priors *implicitly*.

**Stage 1: Global Structure and Distortion Learning.** In this stage, the model focuses solely on global panorama

generation, aiming to capture the overall 360° scene structure and latitude-dependent distortion patterns. To this end, we train the model on both *Text-to-Panorama* (T2P), which input is a textual description, and *Image-to-Panorama* (I2P), which inputs include both a perspective image and a corresponding text prompt. The standard denoising loss $\mathcal{L}_{\text{denoising}}$ is applied to reconstruct the panorama.

**Stage 2: Localized Controllable Generation.** In this stage, the model is transitioned to fine-grained, object-level control. To achieve this, we train the model to synthesize new objects under explicit spatial (*e.g.*, via bounding boxes) and appearance (*e.g.*, via reference images) constraints. Specifically, given a source panorama $I_{\text{src}}$ and control signals $\mathcal{C}_{\text{gen}} = \{P_{\text{gen}}, B, I_{\text{ref}}\}$, the model $G$ generates a new panorama $I_{\text{tgt}}$ with the specified objects inserted. During training, both the layered shape loss $\mathcal{L}_{\text{shape}}$ and the denoising loss $\mathcal{L}_{\text{denoising}}$ are applied. The resulting controllable generation model is used as a data generator for Stage 3.

**Stage 3: Supervised Editing Learning.** Building on the controllable generation model $G$ from Stage 2 and paired data construction in Sec. 3.3, we synthesize triplets $(I_{\text{src}}, \mathcal{P}_{edit}, I_{\text{tgt}})$ and train the final editing model $E$ on these instruction-driven pairs. During training, we apply both layered shape loss $\mathcal{L}_{\text{shape}}$ and the denoising loss $\mathcal{L}_{\text{denoising}}$.

**Progressive Task Transition.** To prevent instability from abrupt task switching, we gradually transition training across stages. Data from the previous stage is linearly decayed as new-stage data increases, while a small portion of earlier data is retained to preserve prior capabilities. This strategy enables continual learning without forgetting, and yields a unified model covering all tasks.

### 3.3. ERP-Native Data Synthesis

To enable diverse panoramic editing, we first require a controllable generation model that can flexibly synthesize new content under user-defined conditions.

#### 3.3.1. CONTROLLABLE PANORAMA GENERATOR

An ideal generator should satisfy three criteria: (i) support multiple forms of control; (ii) handle multiple objects with coherent geometry; and (iii) produce distortion-consistent panoramas directly in the ERP domain.

Formally, we train a panoramic generation model $G$ that takes as input a source panorama $I_{\text{src}}$ and an external condition $C_{\text{gen}}$, and outputs a synthesized panorama $I_{\text{tgt}} = G(I_{\text{src}}, C_{\text{gen}})$, where the condition $C_{\text{gen}}$ provides both semantic and spatial guidance, and is defined as: $C_{\text{gen}} = \{P_{\text{gen}}, \{(B_k, I_{\text{ref}}^k)\}_{k=1}^{K}\}$, where $P_{\text{gen}}$ is a text prompt describing the desired panoramic scene, and each tuple $(B_k, I_{\text{ref}}^k)$ denotes the $k$-th object's spatial region specified by a bound-

ing box $B_k$ and its corresponding reference image (optional). During inference, we encode the source panorama $I_{\text{src}}$, and each reference image $I_{\text{ref}}^k$ into latent features using the VAE encoder. Each bounding box $B_k$ is downsampled to match the latent resolution, and the resulting spatial masks are concatenated with the corresponding image and text tokens in a fixed order to form the conditioning input. During training, we selectively drop elements from $C_{gen}$ to simulate diverse control scenarios. This stochastic conditioning encourages the model to adapt to varying levels of guidance, thereby enhancing its generalization and robustness.

### 3.3.2. PAIRED DATA CONSTRUCTION

Build on the above model $G$, we construct a dataset $D = \{(I_{\text{src}}, P_{edit}, I_{\text{tgt}})_i\}_{i=1}^N$, where each triplet consists of a source panorama $I_{\text{src}}$, an editing instruction $\mathcal{P}_{edit}$, and the edited target panorama $I_{\text{tgt}}$. For each $I_{\text{src}}$, we synthesize multiple $I_{\text{tgt}}$ associated with distinct editing operations, covering five representative types:

- **Addition.** We randomly sample several bounding boxes $B$ as potential insertion regions. For each $B$, GPT-5 (OpenAI, 2025) first generates a suitable object description; we then retrieve a matching reference image $I_{\text{ref}}$ from the internet and ask GPT-5 again to produce a global text prompt $P_{\text{gen}}$. Finally, the target panorama $I_{tgt}$ is synthesized via $G(I_{\text{src}}, \{P_{\text{gen}}, \{B, I_{\text{ref}}\}\})$.

- **Removal.** This is obtained by swapping the source ($I_{\text{src}}$) and target ($I_{tgt}$) images from the addition pairs.

- **Replacement.** It is generated by inserting two different objects at the same location $B$ within one $I_{\text{src}}$. The two generated panoramas form a replacement pair.

- **Movement.** It is produced by placing the same reference object $I_{\text{ref}}$ at different target locations $B_1$ and $B_2$.

- **Modification.** It is created by applying two addition operations at the same position using different references $I_{\text{ref}}^{(1)}$ and $I_{\text{ref}}^{(2)}$, where $I_{\text{ref}}^{(2)}$ is obtained via the editing model (Batifol et al., 2025b). For global changes, $G$ performs scene-level edits from text prompts.

With constructed data, GPT-5 automatically compares each $(I_{\text{src}}, I_{\text{tgt}})$ pair and generates the corresponding instruction $P_{\text{edit}}$, resulting in a large-scale, diverse dataset for supervised training. Finally, to further ensure data quality, trained annotators are asked to lightly validate a subset of pairs.

## 4. Experiments

### 4.1. Evaluation Protocol

#### 4.1.1. SETUP

We adopt Qwen-Image-Edit-2509 (Wu et al., 2025a) as the base model for panorama generation and editing, and fine-

tune it using LoRA (Hu et al., 2022) with rank 32, applied to the attention layers. By default, all panoramic images are trained at a resolution of $512 \times 1024$. All stages are trained on 8 NVIDIA A100 GPUs with a per-GPU batch size of 4, using AdamW with a learning rate of $1 \times 10^{-4}$. Stage 1 is trained for 7k steps (8 hours), while stages 2 and 3 are each trained for 10k steps, taking approximately 22 hours and 18 hours, respectively. Training data for stages 1 and 2 is sourced from Sun360 (Xiao et al., 2012), Structured3D (Zheng et al., 2020), and Pano360 (Kocabas et al., 2021), along with additional images rendered from UE scenes. The UE scenes are also used to render the paired training data for stage 3. To enable high-resolution editing (*e.g.*, 2K), we discard the layering-related input/output components and fine-tune the model for 1500 additional iterations.

#### 4.1.2. DATASET

We present **PEBench**, a comprehensive panorama editing benchmark that includes both real-world and synthetic panoramas, covering indoor and outdoor scenes. **PEBench** consists of 200 panoramas (100 real-world and 100 synthetic) designed to evaluate the editing performance across diverse environments.

- *Real-World Data:* We have collected 100 panoramas from online sources (*e.g.*, Freepik). To ensure diversity, GPT-5 (OpenAI, 2025) is used to generate 20 distinct search queries (10 indoor and 10 outdoor). For each query, five high-quality images are manually curated.

- *Synthetic Data:* We have generated 100 panoramas from two complementary sources: (i) Skybox (Skybox, 2025), an AI-based 360° content generator using diverse prompts; (ii) Unreal Engine (Epic Games), rendering panoramas from 3D assets.

To evaluate panorama editing, each test case consists of a (source image, editing prompt) pair. For each panorama, we define five editing tasks: *removal*, *addition*, *replacement*, *movement*, and *modification*. We employ GPT-5 to automatically detect objects and generate task-specific editing instructions conditioned on the identified object categories, yielding 1,000 test pairs (200 images × 5 editing tasks). Refer to Sec. B of Supplemental for more details.

#### 4.1.3. METRICS

We evaluate our framework using a distinct set of metrics for the panorama editing tasks. Following Step1X-Edit (Liu et al., 2025), we adopt the comprehensive *VIEScore* (Ku et al., 2024) to evaluate editing quality from three perspectives: *(1) SC (Semantic Consistency)*: measuring how well the edited image aligns with the given instruction. *(2) PQ (Perceptual Quality)*: assessing visual realism and the presence of artifacts. *(3) O (Overall Score)*: combining the

*Table 1.* Quantitative comparison of our method against eight SOTA image editing methods on **panorama editing**. The user study reports the average ranking in image quality (IQ), distortion accuracy (DA), and text alignment (TA). The best and second-best results are highlighted in **bold** and underlined, respectively. † denotes proprietary methods which cannot be re-trained. All $CLIP_{dir}$ scores are multiplied by 100 for readability.

| Category | Metrics | Qwen (Wu et al., 2025a) | Kontext (Batifol et al., 2025a) | GPT-5† (OpenAI, 2025) | Nano Banana Pro† (Google, 2025) | IC-Edit (Zhang et al., 2025c) | Omni² (Yang et al., 2025b) | Step1X-Edit (Liu et al., 2025) | OmniGen2 (Wu et al., 2025c) | Ours |
|---|---|---|---|---|---|---|---|---|---|---|
| Automatic | FID ↓ | 65.42 | 60.18 | 92.37 | 51.10 | 84.83 | 61.15 | 78.69 | 66.53 | **45.25** |
| | $CLIP_{dir}$ ↑ | 14.57 | 13.05 | 12.92 | 16.74 | 16.21 | 16.65 | 11.93 | 13.24 | **19.62** |
| | SC ↑ | 5.91 | 3.62 | 2.53 | 5.88 | 5.74 | 5.86 | 5.41 | 5.98 | **8.15** |
| | PQ ↑ | 5.42 | 4.03 | 4.91 | 5.41 | 5.23 | 5.36 | 5.48 | 5.11 | **7.87** |
| | O ↑ | 5.66 | 2.77 | 2.73 | 5.64 | 5.48 | 5.60 | 5.44 | 5.53 | **8.01** |
| User study | IQ ↓ | 4.58 | 7.23 | 8.91 | 2.37 | 6.76 | 5.05 | 5.77 | 3.14 | **1.19** |
| | DA ↓ | 2.83 | 6.94 | 8.78 | 3.41 | 7.07 | 5.22 | 5.34 | 4.15 | **1.26** |
| | TA ↓ | 3.31 | 7.52 | 8.83 | 2.18 | 4.95 | 5.09 | 8.02 | 3.76 | **1.34** |

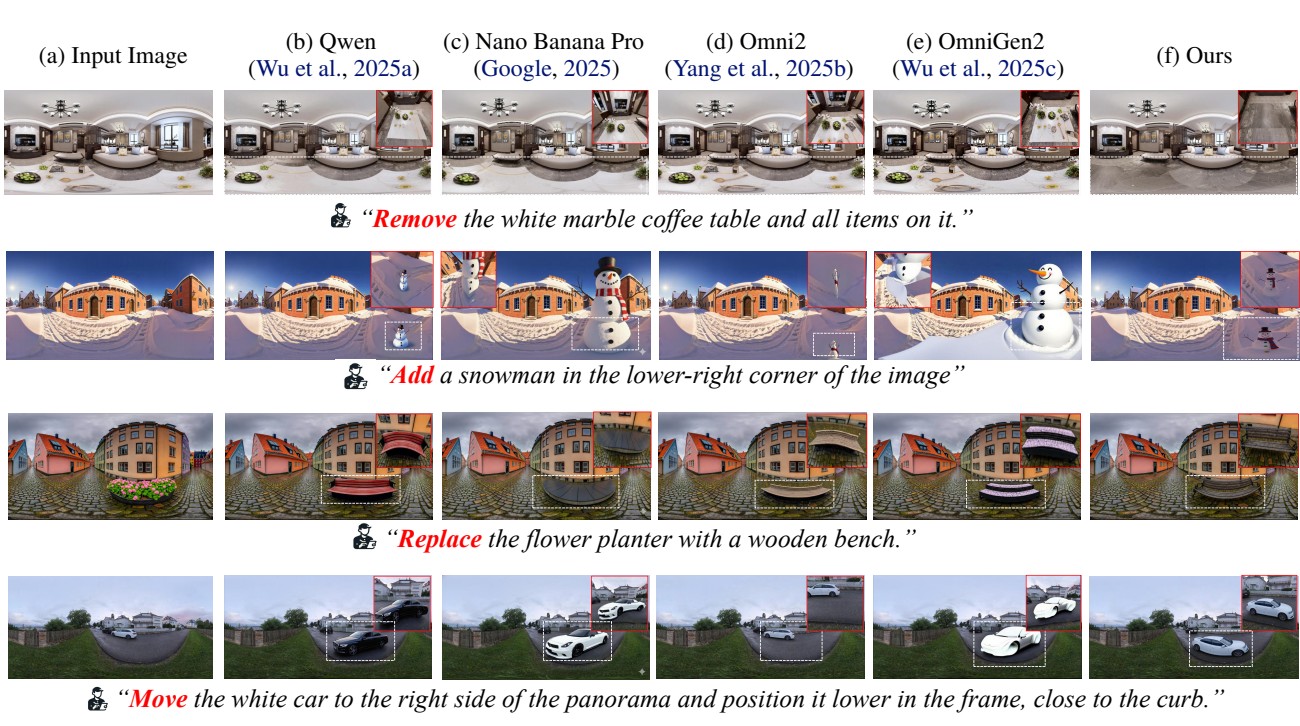

| (a) Input Image | (b) Qwen (Wu et al., 2025a) | (c) Nano Banana Pro (Google, 2025) | (d) Omni2 (Yang et al., 2025b) | (e) OmniGen2 (Wu et al., 2025c) | (f) Ours |

*"**Remove** the white marble coffee table and all items on it."*

*"**Add** a snowman in the lower-right corner of the image"*

*"**Replace** the flower planter with a wooden bench."*

*"**Move** the white car to the right side of the panorama and position it lower in the frame, close to the curb."*

*"**Modify** the wooden walkway into a stone walkway and extend the walkway forward."*

*Figure 4.* Qualitative comparison with four top-performing SOTA methods from Tab. 1 on panorama editing. White boxes indicate the selected regions for visualization. The corresponding edited areas are shown in the perspective view within the red boxes. Refer to Sec. D.6 of Supplemental for more results.

above two aspects as a unified score. Additionally, we also introduce the *Fréchet Inception Distance (FID)* (Heusel et al., 2017) to evaluate the overall visual quality and *CLIP Direction Score (Shi et al., 2024)* to measure consistency between the edited image and the instruction.

### 4.2. Comparisons with SOTAs

We compare ***World-Shaper*** with 8 SOTA editing methods: Qwen-Image-Edit-2509 (Wu et al., 2025a), FLUX.1 Kontext [dev] (Batifol et al., 2025a), GPT-5 (OpenAI, 2025), Nano Banana Pro (Google, 2025), OmniGen2 (Wu et al., 2025c),

Omni2̂ (Yang et al., 2025b), Step1X-Edit (Liu et al., 2025), and IC-Edit (Zhang et al., 2025c). For fairness, we retrain all methods with available training codes.

**Quantitative Comparison.** Tab. 1 presents the quantitative comparison. Our method achieves the best performance on the Overall (O) metric and consistently surpasses all competing approaches across every criterion, demonstrating its comprehensive advantages. Specifically, it attains the lowest FID and highest PQ, indicating superior visual realism and perceptual quality of the generated results. Moreover, the highest $CLIP_{dir}$ and SC further show that our model most

| (a) Input Image | (b) Qwen (Wu et al., 2025a) | (c) Nano Banana Pro (Google, 2025) | (d) OmniGen2 (Wu et al., 2025c) | (e) Ours |

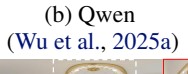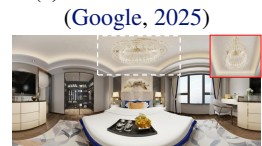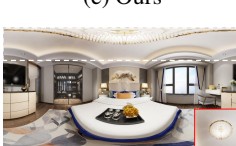
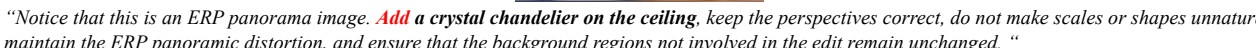

*"Notice that this is an ERP panorama image. **Add a crystal chandelier on the ceiling**, keep the perspectives correct, do not make scales or shapes unnatural, maintain the ERP panoramic distortion, and ensure that the background regions not involved in the edit remain unchanged. "*

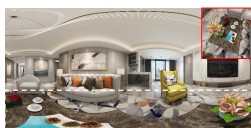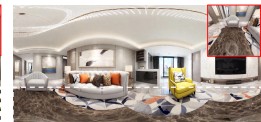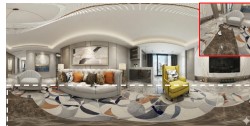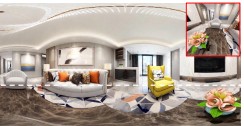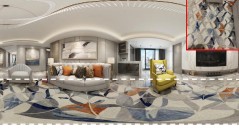

*"Notice that this is an ERP panorama image. **Remove the table and all objects on the table in this image**, keep the perspectives correct, do not make scales or shapes unnatural, maintain the ERP panoramic distortion, and ensure that the background regions not involved in the edit remain unchanged."*

*Figure 5.* Qualitative comparison with 3 top performance methods from Table. 1 using enhanced text prompts.

faithfully adheres to the editing instructions.

**Qualitative Comparison.** Fig. 4 shows the visual comparison. Existing methods exhibit two major limitations. First, **Distortion Unawareness** — they fail to perceive or reason about geometric distortions in panoramic images, leading to various failures depending on the editing type. For *removal* (row 1), distorted objects are misrecognized or ignored (*e.g.*, only part of the objects are removed because the distorted one is not correctly identified). For *addition* and *movement* (rows 2 and 4), newly generated or repositioned objects exhibit unnatural shapes or inconsistent geometry due to incorrect distortion handling. Second, **Background Instability** — some methods, such as Nano Banana Pro and OmniGen2, tend to modify the background regions that should remain unchanged ((c) and (e), row 4). In contrast, our method effectively mitigates both issues by maintaining geometric consistency and precise text-guided control.

**Effect of Prompt Enhancement.** To rule out the possibility that these failures are caused by under-specified prompts, we further evaluate all baselines with enhanced prompts that explicitly indicate the ERP panorama input, encourage preservation of panoramic geometric distortions, and constrain unrelated background regions to remain unchanged. As shown in Fig. 5, prompt enhancement brings only modest improvements, while the core failure modes remain: baselines still struggle with geometrically correct object generation under ERP distortion and distorted-object recognition for removal. This suggests that their limitations mainly stem from the mismatch between perspective-pretrained editing models and panoramic geometry, rather than insufficient prompt wording. Quantitative results and more details are provided in Sec. D.1 of the Supplemental.

**User Study.** We also conduct a user study with 72 participants to evaluate human preferences. For the editing task, each participant is shown 20 cases. Each case contains (1) an input panorama, (2) an editing instruction, and (3) nine

randomly shuffled results from ***World-Shaper*** and the eight competing methods. Participants rank all methods according to three criteria: image quality (IQ), distortion accuracy (DA), and text alignment (TA). We aggregate rankings over all participants and report the averaged scores in the bottom part of Tab. 1. These results show that our method is consistently ranked highest on all three aspects.

### 4.3. Ablation Study

*Table 2.* Addition-based ablation study on adding core components individually to the Qwen baseline. DAAM, LSL, and PCT denote *Distortion-Aware Attention Modulation*, *Layered Shape Loss*, and *Progressive Curriculum Training*, respectively.

| DAAM | LSL | PCT | PQ ↑ | SC ↑ | O ↑ | FID ↓ | CLIP$_{dir}$ ↑ |
|---|---|---|---|---|---|---|---|
| | | | 5.42 | 5.91 | 5.66 | 65.42 | 14.57 |
| ✓ | | | 7.21 | 7.42 | 7.35 | 51.85 | 18.25 |
| | ✓ | | 7.15 | 7.78 | 7.45 | 53.20 | 17.85 |
| | | ✓ | 6.98 | 7.15 | 7.10 | 54.60 | 17.35 |
| ✓ | ✓ | ✓ | **7.87** | **8.15** | **8.01** | **45.25** | **19.62** |

**Distortion-Aware Attention Modulation and Layered Shape Loss.** We first conduct a removal-based ablation on DAAM and LSL, where the progressive curriculum training is enabled by default. Since both Stage 2 and Stage 3 employ shape constraints, we ablate them in Table 3 by removing (i) *both* constraints simultaneously (first row), (ii) the *output-level* layered shape constraint from *both* stages (second row), or (iii) the *feature-level* distortion-aware attention constraint from *both* stages (third row).

Fig. 6 shows the visual result. We draw three main conclusions. ① Using only the denoising loss leads to the poorest performance (Fig. 6(e)), especially in terms of semantic consistency (SC) and overall visual quality (FID). ② Removing either layered shape loss (LSL) or distortion-aware attention modulation (DAAM) leads to noticeable performance deterioration, obvious geometric distortions (Fig. 6(c)) or visual artifacts (Fig. 6(d)), respectively. ③ Combining both

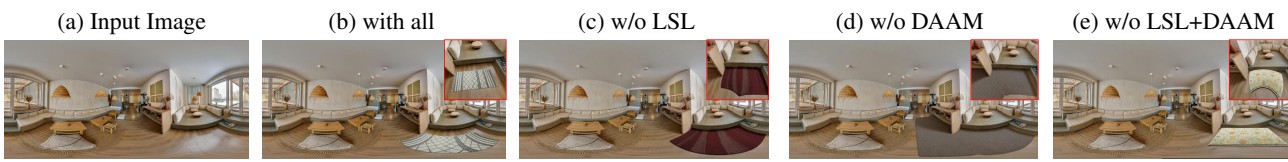

*Figure 6.* Qualitative ablation of Distortion-Aware Attention Modulation and Layered Shape Loss. Editing prompt: "***Add** a carpet to the right side.*"

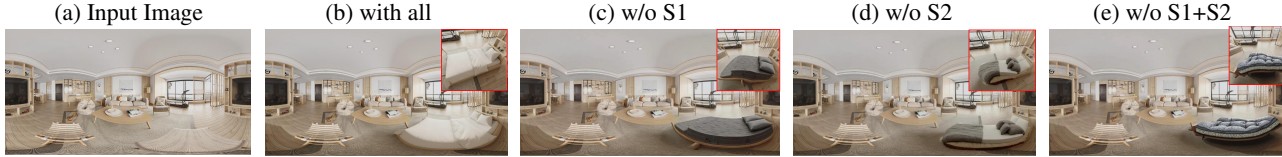

*Figure 7.* Qualitative ablation of Progressive Curriculum Training. Editing prompt: "***Replace** the wooden stool on the right with a bed.*"

components yields complementary benefits, enabling our approach to achieve the best geometric stability and visual quality (Fig. 6(b)).

We further verify their individual effectiveness by adding each component to the Qwen baseline. As shown in Table 2, adding DAAM alone improves O from 5.66 to 7.35, and adding LSL alone improves O from 5.66 to 7.45. These addition-based results are consistent with the removal-based ablation, confirming that both DAAM and LSL independently contribute to geometry-aware panorama editing.

*Table 3.* Ablation study on removing the *Distortion-Aware Attention Modulation* (DAAM) and *Layered Shape Loss* (LSL).

| DAAM | LSL | PQ ↑ | SC ↑ | O ↑ | FID ↓ | CLIP$_{dir}$ ↑ |
|---|---|---|---|---|---|---|
| | | 5.65 | 5.02 | 5.32 | 58.32 | 16.36 |
| ✓ | | 6.92 | 7.84 | 7.45 | 53.82 | 18.04 |
| | ✓ | 6.85 | 7.76 | 7.38 | 54.47 | 17.85 |
| ✓ | ✓ | **7.87** | **8.15** | **8.01** | **45.25** | **19.62** |

**Progressive Curriculum Training.** In this removal-based ablation, DAAM and LSL are enabled by default. We evaluate the editing training (Stage 3) under three settings by removing earlier stages: (i) remove *both* Stage 1 and Stage 2 (i.e., train Stage 3 directly from the Qwen-Image-Edit weights); (ii) remove *only* Stage 2 (i.e., keep only the global generation weights from Stage 1); and (iii) remove *only* Stage 1 (i.e., keep only the localized controllable generation weights from Stage 2).

Fig. 7 and Table 4 show the results. We draw three conclusions. ① Removing Stage 1 (Table 4, row 3) weakens global structural priors, degrading overall geometry and generative metrics (e.g., FID). ② Removing Stage 2 (Table 4, row 2) harms fine-grained controllability, lowering editing-related metrics (e.g., PQ and CLIP$_{dir}$). ③ Eliminating *both* stages (Table 4, row 1) leads to the largest drop across all metrics and incurs the largest distortion, while keeping *both* (Table 4, row 4) achieves the best performance, indicating that Stage 1 and Stage 2 are complementary.

In addition, adding the progressive curriculum training to

the Qwen baseline alone improves O from 5.66 to 7.10, as shown in Table 2. This demonstrates that the progressive curriculum itself provides a strong geometry-aware learning prior, even without explicitly adding DAAM or LSL. Together with the removal-based results, this confirms that the global structure learning stage and localized controllable generation stage are important for robust panorama editing.

*Table 4.* Ablation study on removing key stages of the progressive curriculum training: *Global Structure and Distortion Learning* (S1) and *Localized Controllable Generation* (S2).

| S1 | S2 | PQ ↑ | SC ↑ | O ↑ | FID ↓ | CLIP$_{dir}$ ↑ |
|---|---|---|---|---|---|---|
| | | 6.74 | 7.31 | 7.18 | 56.47 | 17.74 |
| ✓ | | 7.25 | 7.76 | 7.50 | 48.91 | 18.36 |
| | ✓ | 7.02 | 7.68 | 7.43 | 52.33 | 19.11 |
| ✓ | ✓ | **7.87** | **8.15** | **8.01** | **45.25** | **19.62** |

## 5. Conclusion

In this paper, we have presented ***World-Shaper***, a unified framework for panorama editing. We introduced a generate-then-edit paradigm for supervised training, along with a geometry-aware learning strategy for global consistent distortion generation. Extensive experiments on the ***PEBench*** show that our method achieves superior geometric fidelity, editing controllability, and visual realism.

Our work has limitations. When handling object–object interactions, artifacts may appear around the interaction region, as shown in Fig. 8. Future work will incorporate explicit inter-object modeling to enhance physical consistency and extend it to video domain.

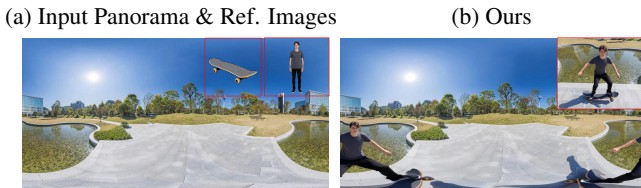

(a) Input Panorama & Ref. Images (b) Ours

🧑 "***Add** a man skateboarding along the edge in front of the pond.*"
*Figure 8.* An example failure case of ***World-Shaper***.

## Impact Statement

This paper presents work whose goal is to advance the field of Machine Learning. There are many potential societal consequences of our work, none which we feel must be specifically highlighted here.

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

# World-Shaper: A Unified Framework for 360° Panoramic Editing
## Appendix

## A. Overview

In this appendix, we provide additional implementation details, ablation studies, and extended evaluations that further support and expand the findings presented in the main paper. We include the following components:

- **Dataset construction and statistics.** Statistical analyses of the *PEBench* and detailed descriptions of the dataset construction procedures used for training our controllable generation and editing models (Sec. B).
- **Model implementation.** Extended implementation details, including full network configurations and the specialized training techniques used in practice (Sec. C).
- **Additional experiments.** Further ablations and extended comparisons for both editing and generation tasks (Sec. D).
- **Application details.** Diverse applications enabled by our method (Sec. E).

## B. Dataset Construction & Analysis

Two distinct data components facilitate our work: (1) a large-scale Training Dataset, essential for the proposed progressive training curriculum, and (2) the challenging Test Benchmark, *PEBench*, necessary for rigorous validation and comparison. This section details the construction methodology and statistical analysis of both.

### B.1. *PEBench* Construction

In the main paper, we briefly introduced our panorama editing benchmark *PEBench*. To comprehensively evaluate our method on panorama generation as well, we extend *PEBench* to cover both panorama editing and panorama generation tasks, including **text-to-panorama** and **image-to-panorama** generation. As shown in Figure 9, the expanded benchmark is constructed through the following steps:

#### B.1.1. PANORAMA COLLECTION

In addition to the 200 high-quality panorama images collected for the panorama editing task, we employed the identical stringent collection standards to acquire a separate set of 200 panorama images dedicated solely to the evaluation of panorama generation methods.

#### B.1.2. CONSTRUCT INPUT DATA FOR PANORAMA GENERATION

We split the panorama generation task into two sub-tasks:

- *Text-to-Panorama:* Each source image is first described by GPT-5 (OpenAI, 2025), which produces a rich, scene-level caption serving as the input text prompt for generation.

- *Image-to-Panorama:* The task's target is to restore the full panorama based on the text prompt and the masked image. Accordingly, each image is partially masked with a randomly generated mask, and GPT-5 (OpenAI, 2025) generates a full-scene description of the original (unmasked) image.

#### B.1.3. CONSTRUCT INPUT DATA FOR PANORAMA EDITING

To evaluate panorama editing, each test case consists of a (source image, editing prompt) pair. We use the collected panorama image as the source image. For each panorama, we define five editing tasks: *removal*, *addition*, *replacement*, *movement*, and *modification*. We employ GPT-5 to automatically detect objects, and then, for each image, generate task-specific editing instructions conditioned on the identified object categories, yielding 1,000 test pairs (200 images × 5 editing tasks).

### B.2. Statistical Analysis of *PEBench*

Our *PEBench* comprises 1,200 test samples spanning both panorama editing and generation tasks, evenly split between real and synthetic sources and between indoor and outdoor scenes. To illustrate its diversity, we provide detailed statistics across five dimensions in Fig. 10(a–e).

- **Image Style Diversity (Fig. 10a):** The benchmark spans six major style categories. Photorealistic images constitute 50% of the dataset, while the remaining 50% covers non-photorealistic domains, including Anime (13%), Watercolor (12%), Low-poly (12%), Fantasy (10%), and Other styles (3%). This balanced composition enables reliable assessment of model generalization and robustness under substantial style and domain shifts.
- **Scene Complexity (Fig. 10b):** Scene complexity is measured based on the density of detectable objects within the panorama. Our benchmark contains scenes of varying complexity: Simple (1–5 objects, 33.8%), **Medium** (6–10 objects, 37.8%), and **Complex** (> 10 objects, 28.5%). This distribution supports robust evaluation across a wide range of scene complexities.
- **Scene Type Coverage (Fig. 10c):** The benchmark spans seven panorama scene types. Indoor scenes include Living/Bedroom (15.8%), Commercial/Office

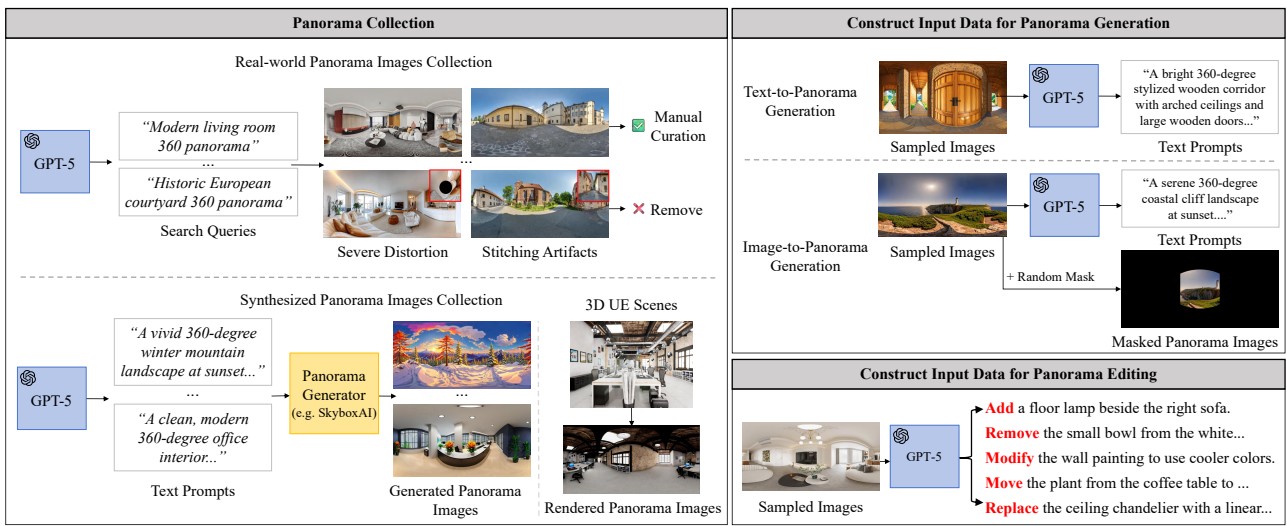

*Figure 9.* Overview of the construction process of our **PEBench**.

(16.2%), and Kitchen/Dining (15.2%), while outdoor scenes cover Nature (17.5%), Urban (15.5%), and Architectural (14.5%). Together with an 'Other' category (5.2%), the dataset maintains a precise 50/50 indoor–outdoor split, supporting reliable evaluation across diverse interior and exterior environments.

- **Complexity of Editing Instructions (Fig. 10d):** We assess the complexity of editing instructions in our benchmark by grouping the editing subset according to the number of targeted objects. Single-object edits represent 54.5%, while multi-object edits constitute a substantial portion (2 objects: 25.0%; 3+ objects: 20.5%). This range supports thorough evaluation across editing instructions of varying complexity.

- **Degrees of Geometric Distortion (Fig. 10e):** We assess geometric distortion in panorama editing by grouping the editing subset according to the latitudinal position of targeted objects. Edits at the equator, where distortion is minimal, account for 33.2%, while the remaining tasks fall in regions with greater geometric distortion (mid-latitude: 35.2%; polar: 31.5%). This balanced distribution enables robust evaluation under varying degrees of geometric distortion.

### B.3. Training Data Construction

To support our progressive three-stage training curriculum, we construct dedicated training datasets for each stage.

#### B.3.1. DATA SOURCES.

Our panorama corpus is assembled from public datasets—Structured3D (Zheng et al., 2020), Pano360 (Kocabas et al., 2021), and SUN360 (Xiao et al., 2012)—and

augmented with custom scenes rendered in Unreal Engine (UE). These sources include both real-world captures and physically based renderings, providing panoramas with faithful equirectangular distortions. We further annotate and process this corpus to produce stage-specific training data.

#### B.3.2. TRAINING DATA FOR GLOBAL GENERATION (STAGE 1).

We prepare data for two tasks focused on global structure and distortion learning:

- *Text-to-Panorama (T2P):* GPT-5 is used to generate descriptive text prompts for each panorama. The text serves as the input, and the original panorama is used as the ground-truth target.

- *Image-to-Panorama (I2P):* We generate perspective images and corresponding masks by randomly projecting each panorama into perspective views. GPT-5 then produces text prompts describing the visible (unmasked) content. Each training sample consists of the perspective image, mask, and text prompt as inputs, with the full panorama as the supervision target.

#### B.3.3. TRAINING DATA FOR LOCALIZED CONTROLLABLE GENERATION (STAGE 2).

Stage 2 focuses on localized controllable generation. Each training sample is represented as a triplet consisting of the source panorama $I_{src}$, the control signals $\mathcal{C}_{gen} = \{P_{gen}, B, I_{ref}\}$, and the supervision targets $(I_{tgt}, L, M)$. The collected raw panoramas serve as ground-truth targets $I_{tgt}$, while all remaining components are derived through a mask-driven pipeline.

- *Object Shape Mask Acquisition:* We extract object masks $M$ from each panorama. For indoor scenes

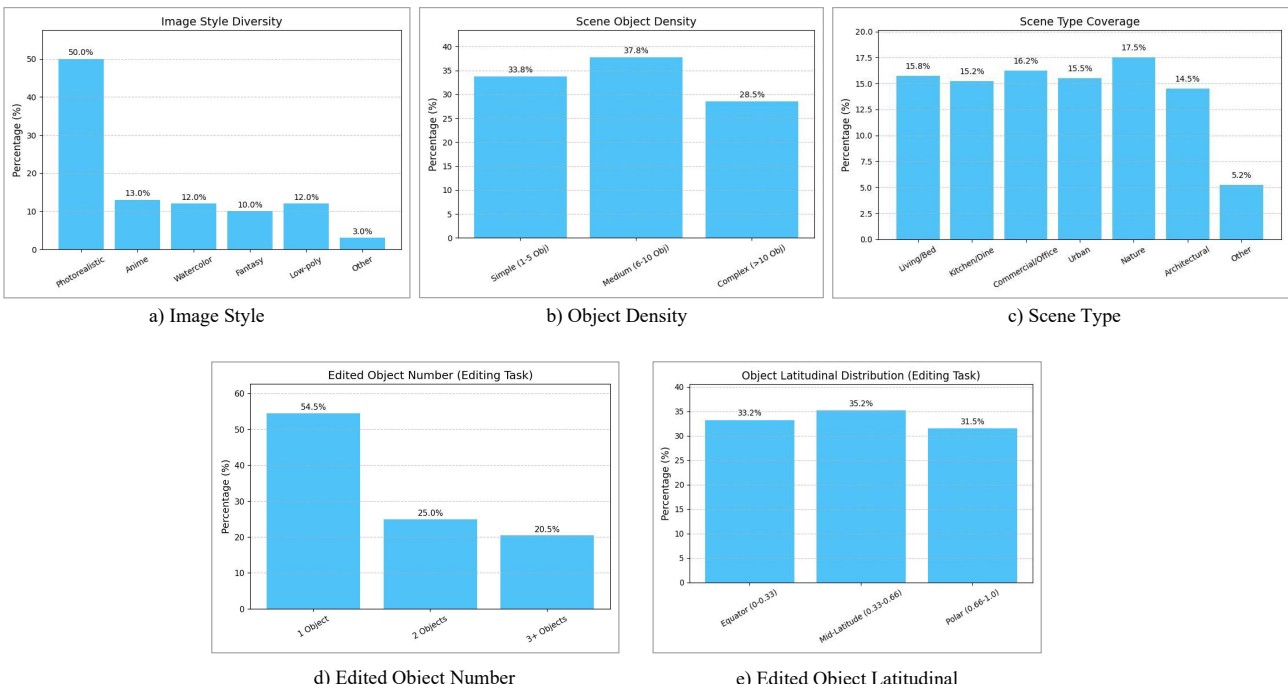

*Figure 10.* Statistical analysis of our ***PEBench*** across five dimensions.

from Structured3D (Zheng et al., 2020), we use the provided semantic labels. For UE-rendered scenes, masks are produced automatically during rendering. For outdoor datasets (Pano360 (Kocabas et al., 2021) and SUN360 (Xiao et al., 2012)), we apply SAM 2 (Ravi et al., 2024) followed by manual refinement.

- *Component Synthesis:* Given $M$, we generate all training components: (1) The **Layer** $L$ is obtained by cropping the masked region of $I_{tgt}$. (2) The **Bounding Box** $B$ is computed from the spatial extent of $M$. (3) The **Reference Image** $I_{ref}$ is created by projecting the masked region into a perspective view. (4) The **Text Prompt** $P_{gen}$ is produced using GPT-5 to describe the global scene context. (5) The **Source Panorama** $I_{src}$ serves as the background. For Structure3D (Zheng et al., 2020), we leverage the provided empty-room panoramas. For UE-rendered scenes, we obtain the background by simply removing the target object from the 3D scene and re-rendering the panorama. For the remaining real-world datasets, we synthesize the background by inpainting the perspective crop defined by $B$ and back-projecting the filled content into the panorama space.

### B.3.4. TRAINING DATA FOR PANORAMA EDITING (STAGE 3).

As described in Sec. 3.3.2, we construct the paired dataset $D = \{(I_{src}, P_{edit}, I_{tgt})\}$ by using the controllable generation model $G$ to synthesize $I_{tgt}$ for various editing operations. In addition to this generative pipeline, we employ a rendering-based approach using Unreal Engine (UE), where edits are applied directly to 3D assets. The modified scenes are re-rendered to produce $I_{tgt}$ with exact geometric consistency. For all collected $(I_{src}, I_{tgt})$ pairs from both pipelines, GPT-5 compares the visual differences and automatically generates the corresponding natural-language instruction $P_{edit}$.

## C. Model Details

### C.1. Details on Progressive Task Transition.

To mitigate the risk of catastrophic forgetting and training instability caused by abrupt task switching, we implement a smooth curriculum transition strategy across the three training stages. Instead of hard cut-offs, we employ a dynamic data mixing protocol:

- **Transition from Stage 1 to 2:** As the training progresses to Localized Controllable Generation (Stage 2), we do not discard the Global Structure data (Stage 1). Instead, we linearly increase the sampling probability of Stage 2 data from $0\%$ to a target ratio (e.g., $80\%$) over the initial epochs, while the remaining $20\%$ consists of Stage 1 data. This ensures that while the model learns fine-grained object control, it retains the fundamental capability of handling global spherical distortions.

- **Transition from Stage 2 to 3:** Similarly, when introducing the Supervised Editing tasks (Stage 3), we maintain containing 20% samples from both Stage 1 (Global) and Stage 2 (Local). This strategy ensures a continuous learning trajectory, where prior capabilities serve as regularization for new tasks, ultimately yielding a unified model proficient in global generation, local control, and instruction-based editing.

## C.2. Training Details

### Stage 1: Global Structure & Distortion Learning.

In implementation, we unify Text-to-Panorama (T2P) and Image-to-Panorama (I2P) by treating T2P as a special case where the mask covers the entire image ($M = 1$). As illustrated in Algorithm 1, given a target panorama $I_{tgt}$, a text prompt $P_{gen}$, and a mask $M$, we first encode the visible context $I_{vis}$ and text prompt $P_{gen}$ using Qwen2.5-VL (Bai et al., 2025), and encode the target panorama $I_{tgt}$ using VAE (Kingma & Welling, 2022). We then construct the network input by concatenating the noisy latent $z_t$, the masked panorama latent $z_{con}$, and the down-sampled mask $m$. Finally, the model is optimized using a standard flow-matching denoising loss.

---

**Algorithm 1** Training Process of Stage 1

**Input:** Target Panorama $I_{tgt}$, Text Prompt $P_{gen}$, Mask $M$
**Output:** Predicted noise $\epsilon_{pred}$

*// Feature Encoding*

$I_{vis} \leftarrow I_{tgt} \odot (1 - M)$
$c \leftarrow \mathcal{E}_{Qwen}(P_{gen}, I_{vis})$
$z_0 \leftarrow \mathcal{E}_{VAE}(I_{tgt})$
$m \leftarrow \text{Downsample}(M)$

*// Noise Injection*

Sample $t \sim \mathcal{U}(1, T)$ and $\epsilon \sim \mathcal{N}(0, I)$
$z_t \leftarrow \text{Scheduler}(z_0, \epsilon, t)$

*// Network Input Construction*

$z_{con} \leftarrow z_0 \odot (1 - m)$
$z_{in} \leftarrow \text{Concat}(z_t, z_{con}, m)$

*// Noise Prediction*

$\epsilon_{pred} \leftarrow \epsilon_\theta(z_{in}, t, c)$

---

### Stage 2: Localized Controllable Generation.

To enforce distortion-aware attention modulation and layered shape loss, we restructure the network output from a single panorama to a composite set containing the full target panorama $I_{tgt}$ and independent layers $\{L_{tgt}^k\}$ for each object.

Given the inputs, we employ Qwen2.5-VL(Bai et al., 2025) and VAE(Kingma & Welling, 2022) to extract text and image features, respectively. We then inject noise into the global and layer targets, constructing the network inputs by concatenating the noisy latents with their corresponding context features, as defined in Algorithm 2.

Considering that processing such multi-layered inputs via simple concatenation hinders effective training due to spatial

---

**Algorithm 2** Training Process of Stage 2

**Input:** Target Panorama $I_{tgt}$, Source Panorama $I_{src}$, Layer Targets $\{L_{tgt}^k\}$, Reference Images $\{I_{ref}^k\}$, Box Masks $\{M_{box}^k\}$, Text Prompt $P_{gen}$
**Output:** Predicted Noise $v_{pred}^{tgt}$, $\{v_{pred}^{(k)}\}$

*// Feature Encoding*

$M_{union} \leftarrow \bigcup M_{box}^k$
$c \leftarrow \mathcal{E}_{Qwen}(P_{gen}, I_{src}, M_{union}, \{I_{ref}^k\})$
$z_{tgt}, z_{src}, \{z_{layer}^k\}, \{z_{ref}^k\} \leftarrow \mathcal{E}_{VAE}(I_{tgt}, I_{src}, \{L_{tgt}^k\}, \{I_{ref}^k\})$
$m_{union}, \{m_{box}^k\} \leftarrow \text{Downsample}(M_{union}, \{M_{box}^k\})$

*// Noise Injection and Input Construction*

Sample $t, \epsilon_{tgt}, \{\epsilon_{layer}^k\}$
$z_t^{tgt} \leftarrow \text{Scheduler}(z_{tgt}, \epsilon_{tgt}, t)$
$\{z_t^k\} \leftarrow \text{Scheduler}(\{z_{layer}^k\}, \{\epsilon_{layer}^k\}, t)$
$z_{in}^0 \leftarrow \text{Concat}(z_t^{tgt}, z_{src} \odot (1 - m_{union}), m_{union})$
$\{z_{in}^k\} \leftarrow \text{Concat}(\{z_t^k\}, \{z_{ref}^k\}, \{m_{box}^k\})$

*// 3D-RoPE ID Assignment*

$S_0 \leftarrow \text{PatchEmbed}(z_{in}^0)$ **with** layer_id $= 0$
**for** each layer $k$ **do**
    $S_k \leftarrow \text{PatchEmbed}(z_{in}^k)$ **with** layer_id $= k$
**end for**
$S_{total} \leftarrow \text{Concat}(S_0, S_1, \ldots, S_N)$

*// Noise Prediction*

$v_{pred}^{tgt}, \{v_{pred}^{(k)}\} \leftarrow \epsilon_\theta(S_{total}, t, c)$

---

ambiguity, we follow ART (Pu et al., 2025) and employ a 3D Rotary Positional Embedding (3D-RoPE) strategy. Distinct from standard 2D RoPE, 3D-RoPE introduces an additional layer_id axis alongside spatial coordinates $(x, y)$. This design enables the self-attention mechanism to efficiently distinguish between tokens belonging to the global canvas (layer_id $= 0$) and specific object layers (layer_id $= k$).

Finally, based on the unified token sequence and global context $c$, the model simultaneously predicts the noise $v_{pred}^{tgt}$ for the global panorama and $\{v_{pred}^{(k)}\}$ for each object layer.

### Stage 3: Supervised Editing Learning.

In the final stage, we convert the generative model into an editing model while preserving the architecture from Stage 2 to maintain consistency and retain previously learned capabilities.

We adapt the editing data to the Stage 2 input format as follows. The source image $I_{src}$ is used as the global context, and the editing instruction $P_{edit}$ serves as the text condition. Because typical editing datasets do not provide explicit object masks or reference images, these signals are replaced with null conditions, implemented analogously to Classifier-Free Guidance: the corresponding tokens or tensors are set to a learnable null embedding or zero tensor. The model's global output target is the edited image $I_{tgt}$. For the object

layers $\{L_k\}$, no ground-truth layers exist in editing pairs, so all target layers are set to transparent.

## C.3. Implementation Details

### C.3.1. HIGH-RESOLUTION ADAPTATION

To extend our framework's capability to high-resolution generation (*e.g.*, $1024 \times 2048$), we introduce a lightweight adaptation phase. During this phase, we disable the auxiliary object layer supervision branches to reduce memory overhead and focus optimization solely on the global view. The model is then fine-tuned on high-resolution panorama images for 1,500 iterations. This strategy effectively preserves the accurate distortion priors learned in earlier stages while refining high-frequency textural details, ensuring the model produces sharp, artifact-free panoramas at higher spatial dimensions.

### C.3.2. EXPLICIT DISTORTION ENCODING

To explicitly encode the distortion of ERP panoramas (with height $H$ and width $W$), which becomes increasingly severe near the poles, we introduce two complementary geometric priors. We first introduce a *Distortion Map* as extra input, $M_D \in \mathbb{R}^{H \times W}$, which encodes latitude-dependent scaling factor and tells the model the intensity of this distortion at each latitude:

$$M_D(x, y) = 1 - D(y) = 1 - \cos\left(\frac{\pi}{2} - \frac{\pi y}{H}\right). \quad (5)$$

Second, following (Xia et al., 2025), we replace the standard i.i.d. Gaussian noise $\mathcal{E} \sim \mathcal{N}(0, \mathbf{I})$ with a *Spatially-Distorted Noise*, $\mathcal{E}_{\text{distorted}}$. To define this noise, we first establish the latitude $\phi(y) = \frac{\pi}{2} - \frac{\pi y}{H}$ for a vertical coordinate $y \in [0, H]$, and the corresponding latitude-dependent scaling factor $D(y) = \cos(\phi(y))$. Based on the principles of spherical projection, the horizontal axis is increasingly stretched as latitude moves from the equator to the poles.

To align our noise distribution with this geometry, we apply this same stretching principle to the noise field. We remap the horizontal sampling coordinate $x$ based on the latitude-dependent scaling factor $D(y)$:

$$x'(x, y) = \frac{W}{2} + \left(x - \frac{W}{2}\right) \cdot D(y). \quad (6)$$

The final $\mathcal{E}_{\text{distorted}}$ is obtained normalized to maintain the distribution on which the diffusion models are pre-trained.

### C.3.3. BOUNDARY-CONSISTENT INFERENCE STRATEGY

Our method performs panorama editing directly in the equirectangular projection (ERP) domain. An important property of ERP panoramas is that they are intrinsically periodic and continuous along the longitudinal (horizontal) direction, where the left and right image boundaries correspond to adjacent regions on the sphere.

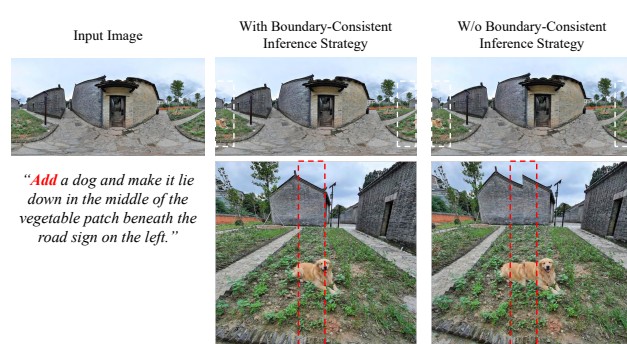

*Figure 11.* Using Boundary-Consistent Inference Strategy during inference time can help improve boundary continuity.

To improve the continuity, we use a *Boundary-Consistent Inference Strategy* that explicitly enforces an approximate periodic boundary condition. As illustrated in Alg. 3, we first perform *Circular boundary extension* by concatenating a narrow strip from the left boundary to the right side in the latent width dimension. This enables the model to access the true neighboring context across the wrap-around boundary during denoising. Next, during the early diffusion steps, we apply a cyclic horizontal shift (*Roll*) to the latent features, which effectively shifts the seam location at the beginning of generation and prevents artifacts from consistently accumulating along a fixed vertical boundary. Finally, we apply *Boundary blending* over the extended band and crop the latent back to the original width, further suppressing pixel-domain boundary artifacts and improving left-right continuity.

## D. Additional Experiments & Results

### D.1. Prompt Enhancement

In Section 4.2, we observe that current baseline methods suffer from two fundamental limitations when applied to ERP panoramic image editing: **Distortion Unawareness** and **Background Instability**. A possible explanation is that these failures may be caused by suboptimal text prompts, since such models heavily rely on language instructions.

To rule out this factor, we further strengthen the prompts by explicitly providing additional cues, encouraging the models to recognize that the input is an *ERP panorama image*, to follow the corresponding geometric distortions, and to preserve the background regions that are not involved in the edit. For example, for the *addition* task, we enhance the prompt as follows: *"Notice that this is an ERP panorama image. Add X to Y, keep the perspectives correct, do not make scales or shapes unnatural, maintain the ERP panoramic distortion, and ensure that the background regions not involved in the edit remain unchanged."*

Fig. 5 shows that even with these prompt enhancements,

*Table 5.* Quantitative comparison of our method against eight SOTA image editing methods using prompt enhancement by GPT-5 (OpenAI, 2025). We report automatic evaluation metrics on PEBench using enhanced prompts for all baselines. The best and second-best results are highlighted in **bold** and underlined, respectively. † denotes proprietary methods which cannot be re-trained. All $CLIP_{dir}$ scores are multiplied by 100 for readability. Numbers indicate the performance after prompt enhancement, with relative improvements over the original prompts shown in parentheses.

| Category | Metrics | Qwen (Wu et al., 2025a) | Kontext (Batifol et al., 2025a) | GPT-5† (OpenAI, 2025) | Nano Banana Pro† (Google, 2025) | IC-Edit (Zhang et al., 2025c) | Omni² (Yang et al., 2025b) | Step1X-Edit (Liu et al., 2025) | OmniGen2 (Wu et al., 2025c) | Ours |
|---|---|---|---|---|---|---|---|---|---|---|
| Automatic | FID ↓ | 63.32 (-2.10) | 58.73 (-1.45) | 88.57 (-3.80) | 50.15 (-0.95) | 82.08 (-2.75) | 59.95 (-1.20) | 75.59 (-3.10) | 64.48 (-2.05) | **44.65** (-0.60) |
| | $CLIP_{dir}$ ↑ | 14.99 (+0.42) | 13.38 (+0.33) | 13.20 (+0.28) | 17.25 (+0.51) | 16.68 (+0.47) | 17.04 (+0.39) | 12.18 (+0.25) | 13.55 (+0.31) | **19.97** (+0.35) |
| | SC ↑ | 6.09 (+0.18) | 3.71 (+0.09) | 2.59 (+0.06) | 6.03 (+0.15) | 5.88 (+0.14) | 6.03 (+0.17) | 5.53 (+0.12) | 6.18 (+0.20) | **8.27** (+0.12) |
| | PQ ↑ | 5.58 (+0.16) | 4.14 (+0.11) | 5.04 (+0.13) | 5.56 (+0.15) | 5.37 (+0.14) | 5.48 (+0.12) | 5.66 (+0.18) | 5.21 (+0.10) | **8.02** (+0.15) |
| | O ↑ | 5.83 (+0.17) | 2.85 (+0.08) | 2.80 (+0.07) | 5.80 (+0.16) | 5.63 (+0.15) | 5.74 (+0.14) | 5.57 (+0.13) | 5.65 (+0.12) | **8.12** (+0.11) |

---

**Algorithm 3** Boundary-Consistent Inference Strategy

**Input:** Source Panorama $I_{src}$, Edit Instruction $P_{edit}$, Model $\epsilon_\theta$, Scheduler $\mathcal{S}$, VAE Encoder/Decoder $\mathcal{E}_{VAE}, \mathcal{D}_{VAE}$

**Hyper-params:** Boundary extension width $b$, Shifting offset $s$, Shift steps $K$, Denoise steps $T$

**Output:** Boundary-consistent Edited Panorama $\hat{I_{tgt}}$

        // Encode panorama into latent space

$z_0 \leftarrow \mathcal{E}_{VAE}(I)$

        // Circular boundary extension

$\tilde{z}_0 \leftarrow \text{Concat}(z_0, \ z_0[:,:,:,1{:}b])$

        // Initialize noisy latent

Sample $\epsilon \sim \mathcal{N}(0, I)$

$\tilde{z}_T \leftarrow \mathcal{S}(\tilde{z}_0, \epsilon, T)$

        // Denoising loop with early-stage shifting

**for** $t = T, T-1, \ldots, 1$ **do**

    $\epsilon_{pred} \leftarrow \epsilon_\theta(\tilde{z}_t, t, P_{edit})$

    $\tilde{z}_{t-1} \leftarrow \mathcal{S}(\tilde{z}_t, \epsilon_{pred}, t)$

    **if** $t > T - K$ **then**

        // Cyclic shifting for seam suppression

      $\tilde{z}_{t-1} \leftarrow \text{Roll}(\tilde{z}_{t-1}, s, \text{horiz})$

    **end if**

**end for**

        // Boundary blending + crop back

$\tilde{z}_0^\star \leftarrow \text{BlendBoundary}(\tilde{z}_0, b)$

$z_0^\star \leftarrow \tilde{z}_0^\star[:,:,:,1{:}W]$

        // Decode edited panorama

$\hat{I} \leftarrow \mathcal{D}_{VAE}(z_0^\star)$

---

cate the target object within a specific cube face, apply a perspective image editing model on that cube, and then project the edited result back to the ERP panorama, this cube-based pipeline suffers from several inherent limitations.

**Difficulty in editing cross-face objects.** First, panoramic scenes often contain objects that span a wide field of view. As illustrated in Fig. 12 (b), when the target object extends across multiple cube faces, it becomes difficult to generate satisfactory results. In such cases, the object must be edited separately on several cube views (e.g., left, front, and right). This fragments the object into multiple parts, breaking its geometric continuity. Consequently, noticeable artifacts frequently appear at cube boundaries when the edited faces are stitched back together (Fig. 12 (b) ). In contrast, our ERP-based editing framework operates directly on the continuous panoramic representation, avoiding such segmentation artifacts.

**Challenges in global style editing.** Second, cube-based approaches are not well suited for global style editing. Since the panorama is processed as disjoint perspective views, enforcing consistent appearance changes across the entire scene becomes challenging. Our ERP-based method naturally supports global style edits over the full panoramic image, as demonstrated in Fig. 15.

**Complicated multi-stage pipeline.** Third, cube-based editing pipelines involve a complicated multi-stage procedure. They require decomposing the panorama into cube faces, selecting the relevant views, performing edits on each face separately, and finally projecting and stitching the results back to the ERP domain. This process becomes even more cumbersome for multi-object editing, as multiple faces may need to be edited sequentially. In contrast, our ERP-based method supports editing multiple objects directly in a single forward pass, avoiding repeated decomposition and reprojection steps.

To further validate the advantage of directly operating on ERP panoramas, we conduct an additional experiment by decomposing each ERP training image into six cube faces, and retraining the baseline methods using cube inputs instead of ERP images.

existing baseline methods still fail to achieve correct edits. Furthermore, we also adopt an LLM (e.g., GPT) to automatically enhance the prompts by injecting additional cues, such as explicitly indicating that the input is a panoramic image, that the ERP distortion should be preserved, and that background regions unrelated to the editing target should remain unchanged. We then re-evaluate these baselines on ***PEBench***. As shown in Table 5, the reported results are obtained using enhanced prompts, with the numbers in parentheses indicating the relative improvement over standard prompts. While prompt enhancement can moderately improve the performance of baseline models, generating geometrically correct distortions remains challenging for these methods.

**D.2. Advantages of Using ERP-native Representation**

One simple workaround for panorama editing is to first lo-

*"**Add** a continuous safety fence around the edge of the lakeside walkway."*

*Figure 12.* When editing objects spanning multiple cube faces, cube-map based pipelines may introduce noticeable artifacts at face boundaries.

*Table 6.* Comparison between ERP and cube-map inputs for representative editing baselines on **panorama editing**. The best results are highlighted in **bold**.

| Method | FID ↓ | CLIP$_{dir}$ ↑ | SC ↑ | PQ ↑ | O ↑ |
|---|---|---|---|---|---|
| Qwen(Wu et al., 2025b) (ERP Input) | 65.42 | 14.57 | 5.91 | 5.42 | 5.66 |
| Qwen(Wu et al., 2025b) (Cube-map Input) | 73.92 | 12.82 | 5.14 | 4.66 | 4.98 |
| Kontext(Batifol et al., 2025b) (ERP Input) | 60.18 | 13.05 | 3.62 | 4.03 | 2.77 |
| Kontext(Batifol et al., 2025b) (Cube-map Input) | 68.61 | 11.22 | 3.22 | 3.51 | 2.35 |
| OmniGen2(Wu et al., 2025c) (ERP Input) | 66.53 | 13.24 | 5.98 | 5.11 | 5.53 |
| OmniGen2(Wu et al., 2025c) (Cube-map Input) | 75.18 | 11.52 | 5.14 | 4.50 | 4.76 |
| Ours (ERP Input) | **45.25** | **19.62** | **8.15** | **7.87** | **8.01** |

As shown in Table. 6, models trained with ERP inputs consistently achieve better performance than those trained with cubemap inputs across all evaluation metrics. This suggests that ERP inputs may better align with the priors of pretrained editing models. In contrast, cubemap-based training could introduce asymmetric face-wise distortions and discontinuities, which may disrupt spatial consistency and increase optimization difficulty.

### D.3. Comparisons on Panoramic Image Generation

We further evaluate our method on panorama generation. For text-to-panorama generation, we compare *World-Shaper* with 6 SOTA methods: UniPano (Ni et al., 2025), Panfusion (Zhang et al., 2024), SMGD (Sun et al., 2025), Diffusion360 (Feng et al., 2023), MVDiffusion (Tang et al., 2023) and DiT360(Feng et al., 2025).

For image-to-panorama generation: we compare *World-Shaper* with 6 SOTA methods: against PanoDiff(Wang et al., 2023), OmniX (Huang et al., 2025a), DreamCube(Huang et al., 2025b), CubeDiff(Kalischek et al., 2025), PanoDiffusion (Wu et al., 2023), and OmniDreamer (Akimoto et al., 2022).

**Generation Metrics:** We employ Fréchet Inception Distance (FID) (Heusel et al., 2017), Inception Score (IS) (Salimans et al., 2016), and Fréchet Auto-Encoder Distance (FAED) (Oh et al., 2022; Zhang et al., 2024) to measure realism, diversity and geometric consistency. Again, CLIP-Score is also utilized to measure the text-image alignment.

**Quantitative Comparisons.** Table 7 and Table 8 reports quantitative results for both text-to-panorama and image-to-panorama generation. Our method consistently outperforms all baselines across automatic metrics and user evaluation.

**Qualitative Comparison.** We also provide visual comparisons in Fig. 13 and Fig. 14. Existing methods either fail at text–image alignment, or exhibit noticeable geometric inconsistencies, including edge artifacts and distortions. Our method avoids these issues, delivering realistic, distortion-consistent, and instruction-aligned results.

**User Study.** We further conduct a user study with 65 participants to assess human preferences across both text-to-panorama and image-to-panorama generation. Each participant is shown 20 text-to-panorama cases (a text prompt with shuffled results from all competing methods) and 20 image-to-panorama cases (a masked input panorama with shuffled outputs). For each case, participants rank all methods along three criteria: image quality (IQ), distortion accuracy (DA), and text alignment (TA). We aggregate rankings across all participants and report the averaged scores in the bottom rows of Table 7 and Table 8. These results show that our method is consistently ranked highest on all three aspects.

### D.4. More Evaluation on Public Available Datasets

To provide a more comprehensive and convincing evaluation besides *PEBench*, we further conduct experiments on

*Table 7.* Quantitative comparison against SOTA methods on **Text-to-Panorama** generation. The user study reports the average ranking in image quality (IQ), distortion accuracy (DA), and text alignment (TA). The best and second-best results are highlighted in **bold** and underlined, respectively.

| Category | Metrics | UniPano (Ni et al., 2025) | Panfusion (Zhang et al., 2024) | SMGD (Sun et al., 2025) | Diff360 (Feng et al., 2023) | MVDiff (Tang et al., 2023) | DiT360 (Feng et al., 2025) | **Ours** |
|---|---|---|---|---|---|---|---|---|
| Automatic | FID ↓ | 57.53 | 81.01 | 69.25 | 88.09 | 96.07 | 51.42 | **48.39** |
| | CLIP-Score ↑ | 28.03 | 28.10 | 28.81 | 27.59 | 27.71 | 28.45 | **28.89** |
| | IS ↑ | 4.02 | 3.91 | 3.95 | 3.35 | 3.13 | 4.15 | **4.45** |
| | FAED ↓ | 8.15 | 9.44 | 9.02 | 11.22 | 15.35 | 8.35 | **7.21** |
| User study | IQ ↓ | 3.57 | 5.14 | 4.88 | 5.21 | 5.65 | 2.42 | **1.13** |
| | DA ↓ | 2.62 | 4.73 | 5.68 | 5.09 | 5.74 | 2.98 | **1.16** |
| | TA ↓ | 4.48 | 3.12 | 2.28 | 6.18 | 6.58 | 4.12 | **1.24** |

*Table 8.* Quantitative comparison against SOTA methods on **Image-to-Panorama** generation. The user study reports the average ranking in image quality (IQ), distortion accuracy (DA), and text alignment (TA). The best and second-best results are highlighted in **bold** and underlined, respectively.

| Category | Metrics | CubeDiff (Kalischek et al., 2025) | DreamCube (Huang et al., 2025b) | PanoDiff (Wang et al., 2023) | OmniX (Huang et al., 2025a) | PanoDiffusion (Wu et al., 2023) | OmniDrmr (Akimoto et al., 2022) | **Ours** |
|---|---|---|---|---|---|---|---|---|
| Automatic | FID ↓ | 53.76 | 50.15 | 76.45 | 58.12 | 89.45 | 111.12 | **46.14** |
| | CLIP-Score ↑ | 28.15 | 28.42 | 27.88 | 28.62 | 27.55 | 27.58 | **28.93** |
| | IS ↑ | 3.89 | 4.25 | 3.65 | 4.11 | 3.32 | 2.77 | **4.71** |
| | FAED ↓ | 7.45 | 7.92 | 9.14 | 8.45 | 17.21 | 19.82 | **7.09** |
| User study | IQ ↓ | 3.65 | 2.35 | 5.35 | 2.55 | 6.15 | 6.80 | **1.15** |
| | DA ↓ | 2.22 | 2.32 | 5.25 | 3.83 | 6.35 | 6.85 | **1.18** |
| | TA ↓ | 3.98 | 2.62 | 5.15 | 2.28 | 6.10 | 6.63 | **1.24** |

| (a) SMGD (Sun et al., 2025) | (b) PanFusion (Zhang et al., 2024) | (c) UniPano (Ni et al., 2025) | (d) DiT360 (Feng et al., 2025) | (e) Ours |
|---|---|---|---|---|

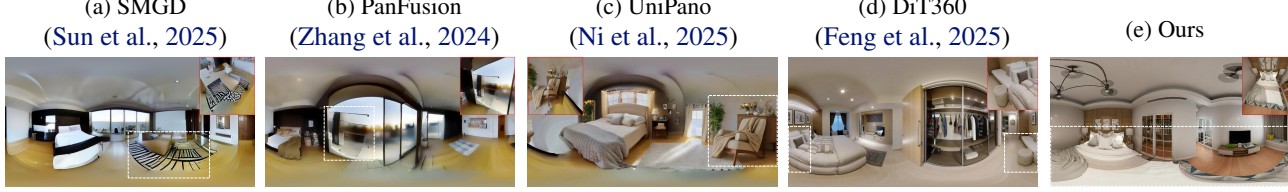

*A modern cozy 360° bedroom with a large layered bed, wooden headboard panels, bedside lamps, a glass-door wardrobe, soft neutral lighting, a TV on a low console, a window nook with cushions, curved ceiling details, and warm wooden flooring, creating a clean comfortable contemporary atmosphere.*

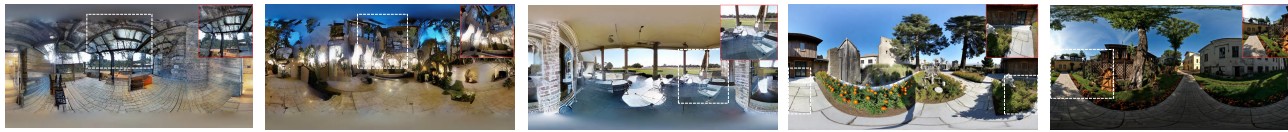

*A sunny 360° courtyard with wooden cottages on the left and an old worn two-story building on the right. Stone pathways curve through green lawns bordered by bright orange flowers. Tall trees cast soft shadows, and rustic wooden garden decorations sit near the cottages. Clear blue sky and warm daylight create a peaceful village atmosphere.*

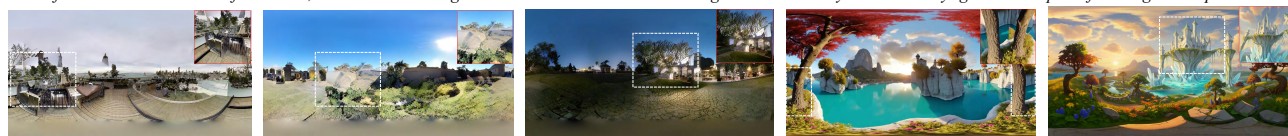

*A bright magical fantasy landscape featuring a floating white castle rising from giant blue crystal formations above a glowing lake. Colorful trees with red, orange, and green foliage frame the scene, surrounded by vibrant flowers and smooth stone paths. Warm sunset light illuminates distant mountains and soft clouds.*

*Figure 13.* Qualitative comparison on Text-to-Panorama generation with top performance methods in Tab. 7.

several publicly available datasets. These additional results aim to better demonstrate the generalization ability of our method under diverse settings and larger evaluation scales.

### D.4.1. EVALUATION ON THE PUBLIC TEST SET OF CONCURRENT WORK

First, we evaluate our method on the public test set released by concurrent work SE360 (Zhong et al., 2025), which contains **660** test cases. Since this dataset is constructed independently and is publicly accessible, it serves as a reliable

(a) Input Masked Panorama | (b) CubeDiff. (Kalischek et al., 2025) | (c) OmniX (Huang et al., 2025a) | (d) DreamCube (Huang et al., 2025b) | (e) Ours

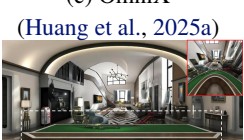 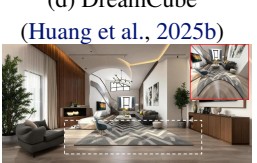 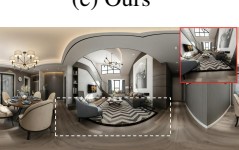

*A modern 360° open-plan living and dining room with neutral tones, wood flooring, curved ceiling details, and warm ambient lighting.*

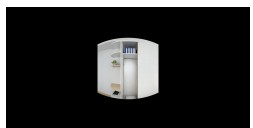 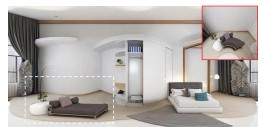 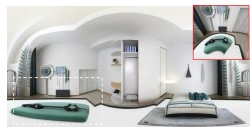 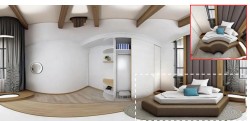 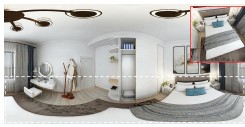

*Modern cozy bedroom, 360 panorama, clean white walls, large soft bed, elegant headboard art, vanity table with mirror, open wardrobe, warm ambient lighting, decorative curtains, soft carpets.*

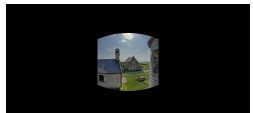 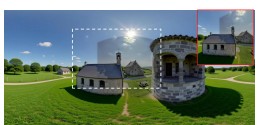 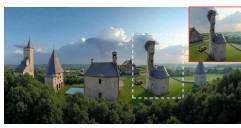 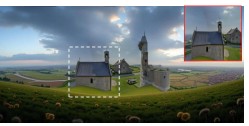 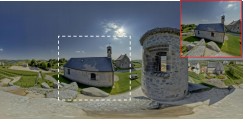

*A bright 360° rural village scene featuring a small stone chapel with a slate roof, an old cylindrical stone lookout tower in the foreground, scattered houses, green fields, and large rocks.*

*Figure 14.* Qualitative comparison Image-to-Panorama generation with top performance methods in Tab. 8.

benchmark for evaluating performance beyond our curated benchmark. As shown in Table 9, our method achieves the best performance on this benchmark across all evaluation metrics.

### D.4.2. LARGE-SCALE EVALUATION ON STRUCTURED3D AND SUN360

To further increase the evaluation scale and reduce potential benchmark-specific bias, we additionally build a large-scale test set based on two widely-used public datasets, **Structured3D** (indoor scenes) and **SUN360** (outdoor panoramic scenes).

While this supplementary benchmark provides a clear advantage in terms of data volume, *PEBench* remains necessary due to its richer editing-oriented design, featuring more diverse scene content (e.g., stylized and UE-rendered panoramas), together with careful manual inspection to ensure the appropriateness and accuracy of the editing instructions.

We randomly sample **1,600** images from each public dataset's **test split**, resulting in a total of **3,200** test images, covering both indoor and outdoor environments, which do not overlap with any training data.

For each image, we generate an editing instruction using GPT, following the same procedure described in Sec. 4.1.2 for *PEBench*. We then evaluate all methods under identical experimental settings on this large-scale benchmark. The results in Tab. 10 show that our approach consistently outperforms strong baselines, confirming that the improvements

generalize beyond small-scale curated benchmarks.

### D.5. Global Style Edit

In addition to supporting local editing and simultaneous multi-object editing for panorama images, our method also facilitates global style editing, as shown in Fig. 15.

### D.6. More Visual Results

We provide more visual results in Fig. 18.

## E. Application

Our method supports diverse applications, and we show the visual results in Fig. 16 and Fig. 17.

***3D world Generation***: Users can begin by generating a panorama from either a text prompt or a local-view input image. A pre-trained depth estimation method (Wang et al., 2025b) is then applied to obtain the corresponding depth map of the panorama. Using this depth information, the 2D pixels are lifted into 3D points, and a sequence of camera poses is defined. The panorama is then rendered along these camera trajectories, and our method is employed to inpaint any missing regions in the rendered views. Finally, a panoramic Gaussian Splatting (GS) representation is optimized using the inpainted panorama frames.

***Indoor Design***: Users fetch a desired piece of furniture from a catalog and specify a location in the room; our method then seamlessly integrates the object into the panoramic scene,

*Table 9.* Quantitative comparison on the **SE360's (Zhong et al., 2025) Benchmark**. Our method is evaluated against eight SOTA image editing baselines for panorama tasks. The best and second-best results are highlighted in **bold** and underlined, respectively. [†] denotes proprietary methods. All $CLIP_{dir}$ scores are scaled ($\times100$) for readability.

| Category | Metrics | Qwen (Wu et al., 2025a) | Kontext (Batifol et al., 2025a) | GPT-5[†] (OpenAI, 2025) | Nano Banana Pro[†] (Google, 2025) | IC-Edit (Zhang et al., 2025c) | SE360 (Zhong et al., 2025) | Step1X-Edit (Liu et al., 2025) | OmniGen2 (Wu et al., 2025c) | Ours |
|---|---|---|---|---|---|---|---|---|---|---|
| | FID ↓ | 67.21 | 58.94 | 88.42 | 49.82 | 81.36 | 62.47 | 76.15 | 64.92 | **43.18** |
| | $CLIP_{dir}$ ↑ | 15.02 | 13.41 | 12.18 | 17.22 | 15.89 | 16.14 | 12.44 | 13.91 | **20.55** |
| Automatic | SC ↑ | 5.64 | 3.48 | 2.62 | 5.92 | 5.51 | 6.02 | 5.12 | 6.14 | **8.43** |
| | PQ ↑ | 5.28 | 4.21 | 4.65 | 5.54 | 5.08 | 5.49 | 5.62 | 4.98 | **8.12** |
| | O ↑ | 5.82 | 2.91 | 2.85 | 5.51 | 5.66 | 5.34 | 5.27 | 5.71 | **8.34** |

*Table 10.* Quantitative results on the combined **Structured3D** (Zheng et al., 2020) and **SUN360** (Xiao et al., 2012) datasets. Our method demonstrates consistent superiority in panorama editing compared to existing SOTA models. The best and second-best results are highlighted in **bold** and underlined, respectively. [†] denotes proprietary methods. $CLIP_{dir}$ scores are scaled by 100.

| Category | Metrics | Qwen (Wu et al., 2025a) | Kontext (Batifol et al., 2025a) | GPT-5[†] (OpenAI, 2025) | Nano Banana Pro[†] (Google, 2025) | IC-Edit (Zhang et al., 2025c) | SE360 (Zhong et al., 2025) | Step1X-Edit (Liu et al., 2025) | OmniGen2 (Wu et al., 2025c) | Ours |
|---|---|---|---|---|---|---|---|---|---|---|
| | FID ↓ | 68.45 | 61.22 | 85.19 | 47.36 | 79.54 | 64.10 | 74.28 | 63.81 | **41.05** |
| | $CLIP_{dir}$ ↑ | 15.68 | 14.12 | 13.05 | 18.41 | 16.32 | 17.02 | 13.56 | 14.23 | **21.37** |
| Automatic | SC ↑ | 5.42 | 3.15 | 2.88 | 6.07 | 5.29 | 6.18 | 5.33 | 6.45 | **8.76** |
| | PQ ↑ | 5.11 | 4.38 | 4.72 | 5.69 | 5.34 | 5.51 | 5.84 | 5.10 | **8.35** |
| | O ↑ | 5.94 | 3.06 | 2.92 | 5.72 | 5.81 | 5.46 | 5.50 | 5.89 | **8.62** |

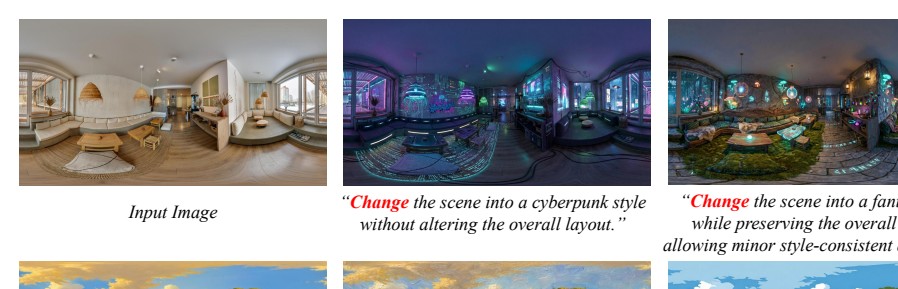
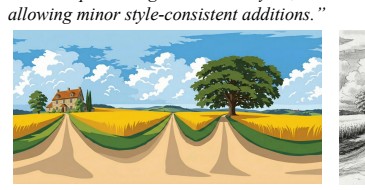
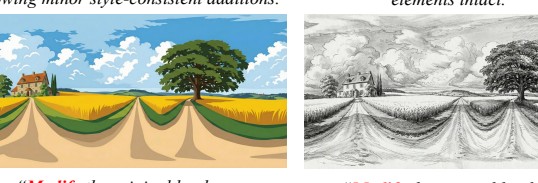

|  |  |  |  |
|---|---|---|---|
| *Input Image* | *"**Change** the scene into a cyberpunk style without altering the overall layout."* | *"**Change** the scene into a fantasy style while preserving the overall layout, allowing minor style-consistent additions."* | *"**Change** the scene to Studio Ghibli Anime Style, keep the location of scene elements intact."* |
| *Input Image* | *"**Modify** the original landscape image into a vivid oil painting style."* | *"**Modify** the original landscape image into a Minimalist Illustration style."* | *"**Modify** the original landscape image into a pencil sketch style."* |

*Figure 15.* Global style editing results of ***World-Shaper***.

adapting to the spherical geometry and lighting conditions for a photorealistic visualization.

(a) Panorama Image      (b) Rendered Views      (c) Panorama Image      (d) Rendered Views

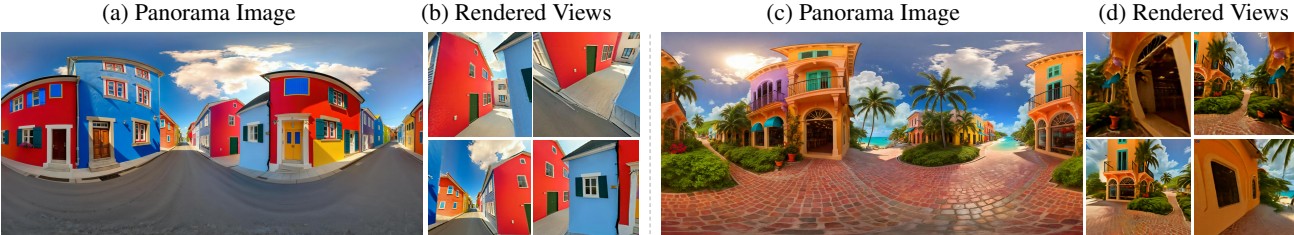

*Figure 16.* In this application, we transform a static panorama image into a 3D world environment. For each scenario, the input panorama image is displayed on the left, while the rendered views from the reconstructed 3D world appear in the four grids on the right.

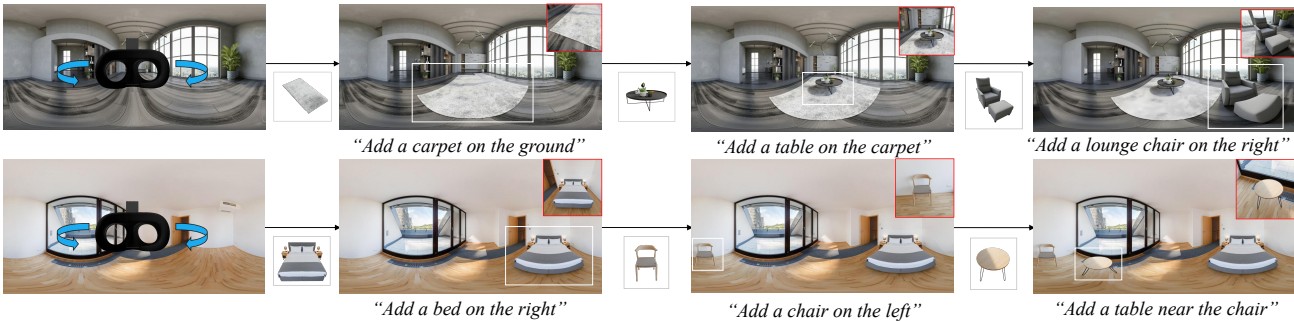

*Figure 17.* In this application, users can experience a fully immersive 360-degree view of interior decoration through a VR headset.

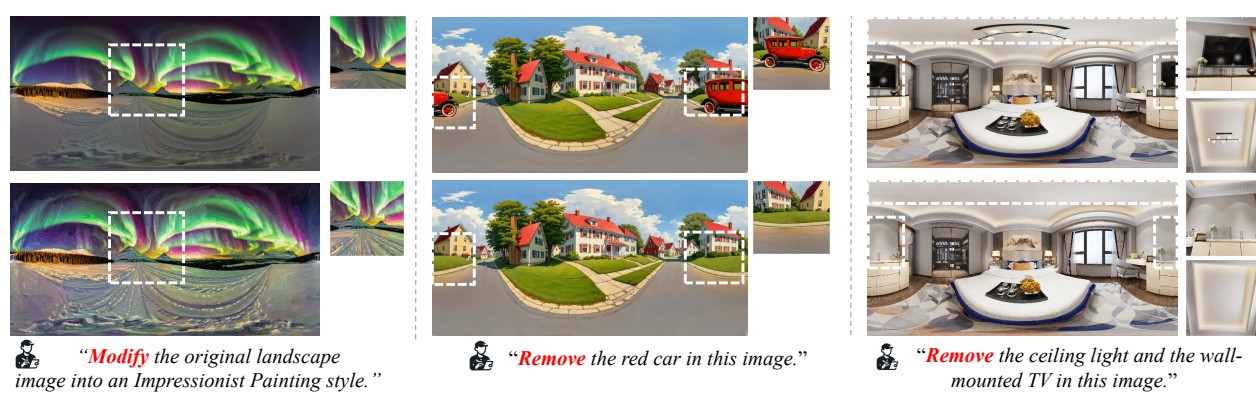

"***Modify*** *the original landscape image into an Impressionist Painting style.*"

"***Remove*** *the red car in this image.*"

"***Remove*** *the ceiling light and the wall-mounted TV in this image.*"

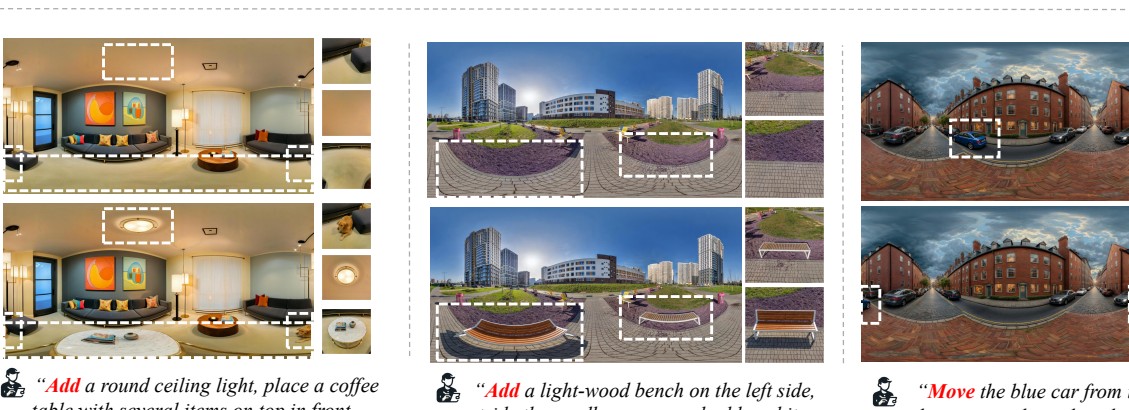

"***Add*** *a round ceiling light, place a coffee table with several items on top in front of the sofa, and include a dog near the right side of the sofa.*"

"***Add*** *a light-wood bench on the left side, outside the sandbox area, and add a white metal-frame bench on the right side, inside the sandbox area.*"

"***Move*** *the blue car from the left side of the street to the right side of the image.*"

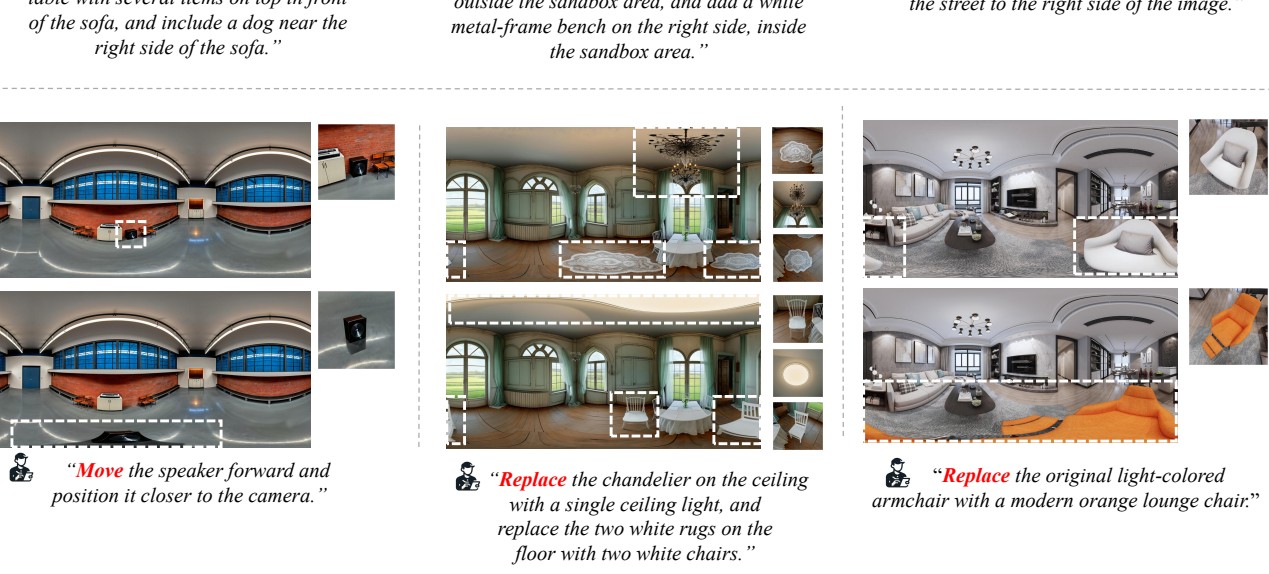

"***Move*** *the speaker forward and position it closer to the camera.*"

"***Replace*** *the chandelier on the ceiling with a single ceiling light, and replace the two white rugs on the floor with two white chairs.*"

"***Replace*** *the original light-colored armchair with a modern orange lounge chair.*"

*Figure 18.* More editing results of our ***World-Shaper*** are shown below. For each case, the corresponding local perspective views are displayed in the right column, with the first row showing the results before editing and the second row showing the results after editing.

