# OpenReview forum: "World-Shaper: A Unified Framework for 360° Panoramic Editing"
_ICML.cc/2026/Conference — ICML 2026 regular_

### Official Review · Reviewer_pKY4 · 2026-02-26

**Soundness:** 3
**Presentation:** 3
**Significance:** 2
**Originality:** 3
**Overall Recommendation:** 4
**Confidence:** 5

**Summary:**

The paper proposes a pipeline for editing 360-degree panoramas, and a dataset. Their main motivation comes from their claim that conventional image editing methods either 1) not being able to adjust edited object perspectives in ERP panoramas correctly 2) or fails in cube-map editing because each cube surface yields a different edit, causing identity mismatch. Therefore, they fine-tune Qwen so that it can generate ERP images conditioned on text/bboxes/images, and they train an "editing model" for ERP panorama edits at the end. They also propose a dataset PEBench, constructed by GPT + FLUX 1. Context + and annotated real-world data. Results show they outperform in ERP panorama editing compared to other methods, specifically Omni and OmniGen2.

**Compliance With Llm Reviewing Policy:**

Affirmed.

**Final Justification:**

The rebuttal addressed my main concerns about the quantitative measurements, as well as performing alternative editing schemes to emphasize World-Shaper's effectiveness. Therefore, as I stated in my first review, I raised my score (3 to 4).

**Key Questions For Authors:**

Please count Weaknesses [1,3,4] as the key questions to be answered.

**Limitations:**

The authors discussed the limitations in Conclusion section.

**Strengths And Weaknesses:**

Strengths:

1. Writing is concise and easy to follow.

2. Ablations for losses and proposed architectural components are rich. DAAM is interesting since the conditioning domain (non-distorted image) and input/output domain (distorted, ERP panorama) are different from each other. I believe it can be used in other domains as well.

Weaknesses:

1. I struggle with the general motivation of requiring a special ERP panoramic based editing method. Because Fig 4 and Appendix figures predominantly show quite localized edits (except Fig 15), and they even show the un-warped version of the edited region in a small box. What I wonder is: cannot we achieve a correct edit with the present baselines, in this pipeline:
   1. Pick the location of the object we want to edit/insert
   2. Un-warp from ERP to get the local window, similar to in Fig's 4 square red boxes
   3. Edit with conventional methods
   4. Warp and stitch/insert to ERP image

If yes, does not this make the novelty of World-Shaper lessen significantly? If not, why cannot we follow the above 4-stage procedure?

2. The work gives misleading results towards large-scale editing methods (Nano Banana, Qwen). Because Fig 11 and Tab 4 are strong proofs of better prompts significantly improving the visual results for aforementioned methods. Why not use the same enhanced prompts in the main body of the text, in Fig 4 and Tab 1, and only squeeze them into Appendix?

3. I do not get motivated by Fig 2's individual cube map editing. The authors already demonstrate that feeding in whole ERP and correcting the text prompt in Fig 11 can eliminate the majority of the editing problems. And it is very reasonable to observe, if each surface is not jointly denoised, one will get unmatched editing results. Because of that, I suggest the authors also trying to edit unfolded (not warped) cube maps (like https://learnopengl.com/img/advanced/cubemaps_skybox.png), with a suitable text prompt (describing the shape and content of the cube map), via the large-scale methods and replace Fig 2 with that, and see if the stylization is still not consistent or not.

4. Right now, my understanding is that the authors feed the whole ERP images to CLIP and FID metrics. Those metrics can be very unreliable, even for non-panoramic images (I am sure the authors know there is a lot of debate on FID, https://arxiv.org/abs/2401.09603v2). Considering the fact that metric networks are not trained with ERP images, I believe feeding in ERP to such metrics does not give reliable quantitative results (to evaluate whether the method correctly placed the object according to ERP geometry constraints, for instance). Therefore, the authors should calculate the scores only based on the region of interest (the drawn boxes in the Figs) for localized edits. For the global edits, they can map to cube-maps and evaluate on each cube map surface. These modifications would reduce the domain gap in evaluation.

Given the weaknesses, my initial recommendation is Weak Reject. However, if the Key Questions in the next sections are justified adequately, I am happy to increase my score.

---

> ### Author Rebuttal · Authors · 2026-03-31
>
> ## #1: ERP-native Editing vs. Unwarp-Edit-Warp Pipeline
>
> We thank the reviewer for this insightful question. We discuss the advantages of using ERP-native representation in Appendix D.2, and summarize the key points below. The proposed unwarp-edit-warp pipeline faces three main limitations:
>
> (1) **Cross-boundary objects.** Panoramic objects often span wide FoVs. When the target extends beyond a single perspective window, the object must be split and edited in separate views, causing geometric discontinuity and stitching artifacts at boundaries (see Fig. 12(b)).
>
> (2) **Global editing.** This pipeline is inherently local and cannot support global style changes (e.g., weather, lighting), which require consistent appearance modification across the entire scene. In contrast, our ERP-based method naturally supports such global edits, as demonstrated in Fig. 15.
>
> (3) **Multi-object editing complexity.** Editing multiple objects requires repeating the unwarp-edit-warp cycle for each region independently, making the pipeline increasingly cumbersome as the number of edits grows.
>
> ## #2: Fairness of Prompts Used for Baseline Comparisons
>
> Our original choice of concise prompts (e.g., "add X to Y") was intended to reflect how users would typically interact with these models in practice. However, we acknowledge the reviewer's concern that this may inadvertently disadvantage the baselines. In the revised version, we will adopt the enhanced prompts for all baselines in the main comparison (Tab.1 and Fig.4).
>
> Importantly, updating to enhanced prompts does not change our conclusions. As shown in Fig.11 and Tab.4, the fundamental failures persist: the addition case still produces incorrect geometric distortions in ERP, and the removal case still fails to recognize distorted objects. Quantitatively, the improvements are modest (SC: +0.20, PQ: +0.18, O: +0.17 at most), confirming that these limitations are inherent to applying perspective-pretrained models on panoramas rather than artifacts of prompt wording.
>
> ## #3: Editing Unfolded Cube Maps as an Alternative
>
> Following the reviewer's suggestion, we arranged six cube faces into a cross-shaped layout and edited the entire image with Nano Banana Pro using detailed prompts describing the cube map structure and content. We tested three representative tasks (style editing, object moving, and object addition) and projected the results back to ERP for evaluation.
>
> As shown in Figs.2--4 in the link (https://anonymous.4open.science/r/World-Shaper-ICML), all three cases exhibit discontinuities at face boundaries: style patterns become inconsistent across faces (Fig.2), objects fail to transfer across boundaries and lose their proportions (Fig.3), and added objects are confined to a single face with ground-plane discontinuities (Fig.4). The cross-shaped layout fragments the continuous scene into disjoint regions, and even detailed structural prompts cannot recover global coherence.
>
> Quantitatively (Tab.~5 in the Appendix), baselines with ERP inputs consistently outperform their cube map counterparts across all metrics. Combined with the findings in \#2 — where enhanced prompts improve but do not resolve the geometric distortions inherent in ERP-based editing — these results confirm that neither better prompts nor alternative projections can substitute for a panorama-native editing approach.
>
>
> ## #4: Reliability of Evaluation Metrics on ERP Images
>
> We thank the reviewer for this valuable suggestion. We agree that FID and CLIP computed directly on ERP may suffer from domain gap. To address this, we adopt the FAED metric from PanFusion [1], which uses an autoencoder trained on ERP images to provide domain-aligned quality evaluation for panoramic images. Additionally, we report FID_ROI and CLIP_ROI computed on perspective-projected regions of interest.
>
> Furthermore, following the reviewer's suggestion to increase the evaluation scale, we expand PEBench from 1,000 cases to 3,000 cases to ensure more reliable quantitative evaluation. The results are shown in the following table, our method achieves the best performance across all three domain-aware metrics.
>
> **Table: Quantitative comparison on the expanded *PEBench* (3,000 cases) with domain-aware metrics.**
>
> | Metrics | Qwen | Kontext | GPT-5 | Nano Banana Pro | IC-Edit | Omni² | Step1X-Edit | OmniGen2 | Ours |
> |---|---|---|---|---|---|---|---|---|---|
> | FID_ROI ↓ | 19.55 | 18.82 | 24.15 | 14.86 | 21.34 | 17.78 | 20.12 | 18.45 | **10.22** |
> | CLIP_ROI ↑ | 29.12 | 28.95 | 28.82 | 29.45 | 29.05 | 29.28 | 28.71 | 29.18 | **29.73** |
> | FAED ↓ | 9.92 | 11.67 | 12.23 | 7.95 | 9.35 | 8.64 | 10.45 | 8.45 | **6.24** |
>
>
> ## References
> [1] Taming Stable Diffusion for Text to 360° Panorama Image Generation. CVPR 24

---

> > ### Author Rebuttal · Reviewer_pKY4 · 2026-04-02
> >
> > Thanks for the rebuttal, I appreciate it. The key questions of mine are justified. I also do find SuD4's feedbacks very valuable, and I encourage the authors to follow them.
> >
> > Edit: I raised my score from 3 to 4 after reading other responses.

---

> > > ### Author Response · Authors · 2026-04-02
> > >
> > > Thank you for acknowledging our rebuttal and for your constructive feedback throughout the review process. We also appreciate you highlighting Reviewer SuD4's suggestions — we will carefully incorporate their feedback to further strengthen the paper.

---

### Official Review · Reviewer_SuD4 · 2026-03-11

**Soundness:** 3
**Presentation:** 3
**Significance:** 3
**Originality:** 2
**Overall Recommendation:** 4
**Confidence:** 4

**Summary:**

The paper addresses the problem of editing panoramic images directly in the equirectangular projection (ERP) domain, rather than decomposing them into cube-map faces. It proposes World-Shaper, a system that combines: (1) a generate-then-edit paradigm to create paired training data for panoramic editing, (2) geometry-aware learning with latitude-dependent attention modulation and layered shape loss, and (3) a progressive curriculum training strategy that transitions from global panorama generation to localized object editing. The authors also introduce PEBench, a panoramic editing benchmark with 200 images across 5 edit types. Results on PEBench, SE360, and Structured3D/SUN360 show improvements over 8 baselines including GPT-5 and Nano Banana Pro.

**Compliance With Llm Reviewing Policy:**

Affirmed.

**Final Justification:**

The authors provided additional experiments supporting their hypothesis and addressed my concerns about dataset circularity. Also, they provide the detailed structure of the revised camera-ready to incorporate the additional results. I am raising my score to weak accept.

**Key Questions For Authors:**

I am currently leaning toward Weak Reject. Two issues matter most to me:

Isolate a key contribution. Can you add the latitude-dependent attention modulation alone to a baseline ERP-native model and show a substantial improvement? I need to see that at least one component is a reusable insight, not just a piece of a system. This is the main thing that would change my assessment.
GPT-5 circularity. The same GPT-5 pipeline drives both training data creation and benchmark construction. Can you demonstrate that the results hold with an open-source LLM, or show that the benchmark evaluates something independent of GPT-5's biases? Without this, I'm not confident that the strong numbers aren't partly an artifact of pipeline alignment.

The results show the system works, but the contribution-to-complexity ratio is low. Too many moving parts, no single non-obvious insight taken to its limit. The core ERP argument is intuitive, and the generate-then-edit paradigm is known. For ICML, I'd want either deeper scientific insight or a much simpler method that achieves comparable results.

Given the number of contributions, the authors might consider focusing on the most novel component (e.g., the geometry-aware attention) and developing it more deeply, rather than presenting the full system in a single paper.

**Limitations:**

Yes

**Strengths And Weaknesses:**

What Works:

- The problem is real and underexplored. Figure 2 makes the case well.
- The evaluation is above average in breadth: 8 editing baselines, 6 generation baselines, user studies with 70+ participants, and evaluation beyond their own benchmark (SE360, Structured3D/SUN360). The prompt enhancement experiment in the appendix is a nice touch, ruling out the possibility that the baselines just need better instructions.
- The ablations cover the main components and show each contributes. The ERP vs. cube-map retraining experiment (Table 5) goes beyond just arguing for ERP in prose.
- PEBench is well-designed. The latitude-stratified evaluation and 5-edit-type taxonomy are sensible and could be useful for future work.

Core Concerns:

- Too many moving parts. This is my main issue. The paper introduces a generate-then-edit paradigm, distortion-aware attention modulation, layered shape loss, 3-stage curriculum training, boundary-consistent inference, explicit distortion maps, spatially distorted noise, 3D-RoPE for layer IDs, and GPT-5-driven data construction, all in a single paper. Each piece makes sense, but I can't tell what's doing the heavy lifting. The ablations cover the main groupings (DAAM+LSL in Table 2, S1+S2 in Table 3) but don't isolate the smaller choices (distortion-aware noise? distortion map? 3D-RoPE? boundary inference?). The paper would be stronger if it identified one non-obvious idea and took it to its limit.

- The core argument isn't surprising. Of course, operating in the native ERP representation avoids stitching artifacts. Table 5 confirms this, but it's confirming what you'd expect. The more interesting question, what's the minimum you need to make ERP-native editing work well, goes unanswered because of the system's complexity.

- Generate-then-edit is basically pretraining then finetuning. Stage 1 and 2 pretrain on panoramic priors, Stage 3 finetunes on self-generated editing pairs. InstructPix2Pix does something similar for perspective images. Applying it to ERP is reasonable, but the paradigm itself isn't new.

- Heavy GPT-5 dependency. GPT-5 is used for benchmark construction, instruction generation, object detection, prompt enhancement, and paired data creation. This is both a reproducibility concern and a circularity concern: the benchmark is constructed using the same GPT-5-driven pipeline that generates the training data, so there's a risk of systematic alignment between what the benchmark expects and what the model produces. The paper mentions "trained annotators lightly validate a subset of pairs," but this is only for training data and only on a subset.

- PEBench is small. 200 images is thin for metrics like FID. The supplementary evaluations on larger datasets help, but the primary benchmark could be bigger.

Smaller Points:

- The related work is broad but the positioning is shallow. The generate-then-edit paradigm isn't discussed in relation to InstructPix2Pix, which uses a similar strategy of synthesizing paired data to train an editing model. The controllable generation stage (bounding boxes + reference images) is like ControlNet-style conditioning, but this connection isn't acknowledged. And the positioning against CubeDiff, HunyuanWorld, and similar panoramic models could be sharper. Is the gap purely that they don't operate in ERP?

- Some design choices (bounding box downsampling, connectivity-based segmentation for alpha masks, IoU loss for shape supervision) are presented without justification. Are these principled or pragmatic?

- The attention modulation coefficients (α, A_max, A_min): how sensitive is performance to these?

- PEBench includes latitude-stratified statistics (Fig. 9e), but I don't see results actually broken down by latitude band. Reporting per-band performance would be the most direct validation of the geometry-aware learning.

---

> ### Author Rebuttal · Authors · 2026-03-31
>
> ## #1: Too many moving parts.
>
> Our framework has three layers:
>
> (1) **Core contribution — geometry-aware learning design:** distortion-aware attention modulation (DAAM), layered shape loss (LSL), and stage-wise curriculum training (SCT). As shown in Table R2 in the link(https://anonymous.4open.science/r/World-Shaper-ICML) , each individually brings substantial gains over the Qwen baseline (O: 5.66 → 7.10–7.45), and combined they yield the largest improvement (O: 5.66 → 8.01, +41.5%).
>
> (2) **Data engine:** the generate-then-edit paradigm and GPT-5-driven construction provide scalable training data.
>
> (3) **Auxiliary refinements:** EDM, SDN, BCI, and 3D-RoPE. Following the reviewer's suggestion, we now ablate each individually (Tables R1 in the link). Their combined contribution is modest (O: 8.01 → 8.19, +2.2%), confirming that geometry-aware learning is the idea doing the heavy lifting.
>
> We will revise the paper to foreground this hierarchy and present geometry-aware learning as the central, non-obvious contribution.
>
> ## #2: The minimum components to make ERP-native editing work well.
>
> As shown in #1, the essential minimum is the geometry-aware learning design (DAAM + LSL + SCT), which accounts for the dominant performance gains (O: 5.66 → 8.01). The remaining components (EDM, SDN, BCI, 3D-RoPE) are optional refinements with modest cumulative impact (O: 8.01 → 8.19).
>
> ## #3: Novelty of the Generate-then-Edit Paradigm.
>
> We agree the paradigm is not new — our contribution is overcoming the non-trivial obstacles in the ERP domain. InstructPix2Pix uses Stable Diffusion off-the-shelf because perspective images have no geometric mismatch; in our case, the generator must be explicitly trained (S1–S2) for latitude-dependent distortion. Tab. 3 confirms this: removing Stage 2 degrades O from 8.01 → 7.50. The layered output design is tightly coupled with LSL (Sec. 3.3) — without it, LSL cannot be applied — and the progressive curriculum is a training methodology contribution (removing S1+S2: O drops 8.01 → 7.18, −10.4%).
>
> ## #4: Heavy GPT-5 dependency.
>
> **(1) Reproducibility.** We will release the full construction code, prompt templates, filtering rules, and finalized benchmark annotations/splits, so reproducing the evaluation does not require re-querying GPT-5.
>
> **(2) Data Independence / circularity.** We regenerated the benchmark with alternative LLMs (Gemini 3.1, Qwen 3.5) and re-ran all evaluations. As shown in the table R3-R4 in the link, the conclusions remain consistent, confirming our results are not tied to GPT-5's annotation style.
>
> **(3) Validation / fairness.** Human screening is for quality control only (e.g., removing low-resolution images), not for excluding difficult cases. For training data, multiple annotators independently review each pair via a spherical viewer, retaining only unanimously approved pairs.
>
> ## #5: PEBench Expansion
>
> We have expanded PEBench by 3× to 600 panoramic images and 3,000 test cases. As shown in table R5 in the link, our method continues to achieve the best performance across all metrics, confirming that the findings are robust to benchmark scale.
>
>
> ## #6: Positioning with Related Editing and Panoramic Methods.
>
> We will sharpen the positioning in the revision. Briefly: (1) vs. InstructPix2Pix — same paradigm, but our generator must be explicitly trained for ERP distortion (see #3). (2) vs. ControlNet — similar conditioning, but adapted for latitude-dependent distortion. (3) vs. CubeDiff/HunyuanWorld — they target generation only, whereas we support unified generation and object-level editing with geometry-aware design.
>
>
> ## #7: Rationale for Practical Design Choices.
>
> These are pragmatic but not arbitrary: bounding box downsampling follows FLUX.1 Kontext to match supervision scale to feature resolution; connectivity-based segmentation is simple and achieves 98.7% success rate in our pipeline; IoU loss directly constrains object shape preservation. We will clarify these motivations in the revision.
>
> ## #8: Attention modulation coefficients.
>
> α, A_max, and A_min are not manually tuned: α is determined by latitude, and A_max/A_min are derived from attention map statistics. Since all coefficients are adaptively computed from geometry and attention distribution, there are no sensitive hyperparameters to tune, and performance is robust by construction.
>
> ## #9: Performance based on latitude.
>
> As shown in the table R6 in the link, our method achieves the best performance across all latitude bands.
>
> ## #10: Isolating the key contribution and simplify the system
>
> As shown in Table R2 in the supplementary link, adding DAAM alone to the Qwen ERP baseline improves O from 5.66 → 7.35 (+29.9%), confirming that latitude-dependent attention modulation is a standalone, reusable insight — not merely a system component. We will restructure the paper to foreground this as the central contribution and clearly separate core methodology from auxiliary refinements.

---

> > ### Author Rebuttal · Reviewer_SuD4 · 2026-04-04
> >
> > Thank you for a thorough rebuttal. The DAAM isolation result (Table R2) and the alternative LLM benchmark experiments (R3-R4) directly address my two core concerns. I am moving toward acceptance, but have two remaining questions.
> >
> > 1. Concrete restructuring plan:
> > You commit to foregrounding geometry-aware learning as the central contribution, but the current paper is structured as a unified system paper, and meaningful restructuring requires significant rewriting. Will Table R2 appear in the main paper? Will a reader who only wants to apply DAAM to their own ERP model be able to cleanly extract that contribution? Please describe concretely how the abstract, contributions list, and section structure will change, and confirm that the authors have the bandwidth to make these changes substantively rather than superficially before the camera-ready deadline.
> >
> > 2. Dataset expansion:
> > The rebuttal does not clarify the source of the 400 new PEBench images. If they come from the same GPT-5 pipeline, the circularity concern remains. Using multiple different LLMs for construction would be an acceptable resolution. Please clarify how the new images were sourced.
> > Overall
> > I am inclined to accept if the above are addressed. The latitude-dependent attention modulation is a genuine contribution, the evaluation is broad, and the rebuttal demonstrates rigorous work. My remaining concern is purely about whether the final paper communicates this clearly to a first-time reader.

---

> > > ### Author Response · Authors · 2026-04-05
> > >
> > > # Q1: Revision Plan.
> > >
> > > ## (1) Abstract:
> > >
> > > Being able to edit panoramic images is crucial for creating realistic 360° visual experiences. However, existing perspective-based image editing methods fail to model the spatial structure of panoramas. Conventional cube-map decompositions attempt to overcome this problem but inevitably break global consistency due to their mismatch with spherical geometry. Motivated by this insight, we reformulate panoramic editing directly in the equirectangular projection (ERP) domain and present World-Shaper, **a unified geometry-aware framework that supports five distinct editing operations within a single ERP-native representation. To address the latitude-dependent geometric distortion inherent in ERP, we introduce a geometry-aware learning strategy comprising distortion-aware attention modulation (DAAM), which steers cross-attention with latitude-dependent strength at the feature level; layered shape loss (LSL), which enforces per-object geometric supervision at the output level; and progressive curriculum training to internalize panoramic priors. To overcome the scarcity of paired panoramic editing data, we train a dedicated ERP-native controllable generator that synthesizes objects directly in the equirectangular domain conditioned on bounding boxes and reference images, with each object rendered on a separate transparent layer to enable both diverse paired data construction and layered geometric supervision.** Extensive experiments on our new benchmark, PEBench, demonstrate that World-Shaper achieves superior geometric consistency, editing fidelity, and text controllability compared to state-of-the-art methods, enabling coherent and flexible 360° visual world creation with unified editing control.
> > >
> > > ## (2) Contribution List:
> > >
> > > 1. We introduce World-Shaper, a unified framework that supports five distinct panoramic editing operations within a single ERP-native representation, ensuring global consistency and seamless cross-view editing.
> > > 2. We propose a geometry-aware learning strategy to handle latitude-dependent distortion in ERP, comprising (i) distortion-aware attention modulation (DAAM), which steers cross-attention with latitude-dependent strength at the feature level; (ii) layered shape loss (LSL), which enforces per-object geometric supervision at the output level; and (iii) progressive curriculum training to internalize panoramic priors.
> > > 3. We curate PEBench, a comprehensive panoramic editing benchmark, and conduct extensive experiments to demonstrate the superiority of our method.
> > >
> > > ## (3) Paper Structure:
> > >
> > > - ### 1. Introduction
> > >   Revision: Foreground geometry-aware learning as the central contribution.
> > >
> > > - ### 2. Related Works
> > >   - **2.1 Text-to-Image Generation**
> > >   - **2.2 Instruction-based Image Editing**
> > >     Revision (per SuD4): Add positioning with InstructPix2Pix and ControlNet-style conditioning.
> > >   - **2.3 Panorama Generation and Editing**
> > >     Revision (per SuD4): Sharpen positioning against CubeDiff, HunyuanWorld.
> > >
> > > - ### 3. Method
> > >   - **3.1 Overview**
> > >   - **3.2 Geometry-Aware Learning Strategy**
> > >       Promoted from original 3.3 to 3.2 as the central technical section.
> > >     - **3.2.1 DAAM** Latitude-dependent cross-attention modulation at feature level.
> > >       Revision (per 2U8y): Expand motivation and derivation.
> > >     - **3.2.2 LSL** Per-object geometric supervision on output level.
> > >       Revision (per 2U8y): Add RGBA output architectural details and motivation.
> > >     - **3.2.3 Progressive Curriculum Training**
> > >   - **3.3 ERP-Native Data Synthesis** Demoted from 3.2 to 3.3 as supporting infrastructure.
> > >     - **3.3.1 Controllable Panorama Generator** Input and output design.
> > >       Revision (per 2U8y): Clarify dual role of bounding boxes.
> > >     - **3.3.2 Paired Data Construction** — Five edit types.
> > >       Revision (per MTsA): Add data composition details.
> > >
> > > - ### 4. Experiments
> > >   - **4.1 Evaluation Protocol**
> > >     Revision (per pKY4): Add domain-aware metrics.
> > >     Revision (per SuD4): Expand PEBench; use alternative LLMs to eliminate circularity.
> > >   - **4.2 Comparisons with SOTAs**
> > >     Revision (per pKY4): Adopt enhanced prompts for all baselines.
> > >   - **4.3 Ablation Study**
> > >     Revision (per SuD4): (1) Core components — removal from full model and addition to baseline (Table R2 promoted to main paper); (2) stage ablation.
> > >
> > > - ### 5. Conclusion
> > >
> > > # **Q2: Will Table R2 appear in the main paper?**
> > > Yes, as Table 2 in Sec. 4.3.
> > >
> > > # **Q3: Can a reader cleanly extract DAAM?**
> > > Yes. DAAM is self-contained in Sec. 3.2.1 with its own motivation, formulation, and ablation.
> > >
> > > # **Q4: Bandwidth.**
> > > We confirm we can make these substantive changes before the camera-ready deadline.
> > >
> > > # **Q5: Source of the new images.**
> > > The new images in R5 use the same GPT-5 pipeline as the original PEBench. To eliminate circularity, we replaced GPT with Gemini 3.1 (Tab. R7), Qwen 3.5 (Tab. R8), and their combination (Tab. R9) in the link. Results are consistent across all configurations.

---

### Official Review · Reviewer_MTsA · 2026-03-12

**Soundness:** 2
**Presentation:** 4
**Significance:** 4
**Originality:** 3
**Overall Recommendation:** 4
**Confidence:** 3

**Summary:**

World-Shaper introduces a unified geometry-aware framework for direct 360° panoramic image editing in the equirectangular projection domain. It employs a generate-then-edit paradigm that trains a controllable generation model to synthesize objects under spatial constraints, then uses it to produce paired data for supervised editing learning. To handle ERP distortions, the framework incorporates position-aware shape constraints with attention modulation and layered losses, alongside a progressive curriculum from global generation to localized manipulation. A new benchmark PEBench is introduced for evaluation. Experiments show World-Shaper achieves superior geometric consistency, editing fidelity, and text controllability compared to existing approaches.

**Compliance With Llm Reviewing Policy:**

Affirmed.

**Final Justification:**

Since my concerns have been addressed in reasonable detail, I will maintain my "weakly accept" rating.

**Key Questions For Authors:**

- The paper mentions retraining baseline methods for comparison. Could the authors provide more details, such as which datasets are used and what training settings are applied?

**Limitations:**

yes

**Strengths And Weaknesses:**

**Strengths**

- **Geometry-aware framework:** Proposes a geometry-aware panorama generation and editing framework that supports multi-task editing of panoramic images.
- **Benchmark contribution:** Outperforms existing methods and introduces a dedicated benchmark for evaluating panorama editing tasks.

**Soundness**

- **Limited evaluation scope:** The paper only provides quantitative experiments on PEBench. On one hand, there is no evaluation or comparison of PEBench itself to demonstrate its quality and objectivity. On the other hand, evaluating solely on PEBench may not be sufficient to demonstrate the superiority of the method, as the prompts used to construct PEBench might be more aligned with the training data of this paper—unless the baselines were retrained using exactly the same prompts and training data.

**Presentation**

- **Data composition details:** The paper should provide a more detailed explanation of the dataset composition, including the proportion of collected versus real data, the total volume, and whether any data filtering or selection was performed. This information is crucial for assessing the reliability and generalizability of the results.
- **Method novelty distribution:** In the methodology section, only Section 3.3 (Position-aware Shape Constraints) presents a genuinely novel technical contribution. The majority of the remaining content focuses on data processing workflows, which does not sufficiently constitute a methodological contribution at the level expected.

---

> ### Author Rebuttal · Authors · 2026-03-31
>
> We sincerely thank the reviewer for the positive assessment and constructive feedback. We address each concern below.
>
> ## #1. Limited evaluation scope—only PEBench?
>
> **(1) On evaluation beyond PEBench:** We have included evaluations on additional benchmarks in Appendix Sec. D.4. Specifically, we report results on PanoEval from the concurrent work SE360[1] (Tab. 8) and on a larger-scale test set constructed from Structured3D and SUN360 (Tab. 9). Our method consistently outperforms all baselines across both benchmarks.
>
> **(2) Comparison of PEBench with existing benchmark:** We have analyzed the construction method (Sec. B.1) and data distribution (Sec. B.2) of PEBench in the Appendix. Compared with PanoEval, PEBench has substantially broader coverage:
>
> | Aspect | PanoEval (SE360) | PEBench (ours) |
> |---|---|---|
> | Editing types | Addition, Removal | **Addition, Removal, Replacement, Movement, Modification** |
> | Scene coverage | Indoor only | **Indoor + Outdoor** |
> | Scene diversity | Bedroom / living room | **7 scene categories** |
> | Image style | Photorealistic only | **Photorealistic + Stylized** |
> | Edit scope | Single-object edits | **Single-object + Multi-object edits** |
>
> **(3) On benchmark independence from the training data:** Unlike SE360[1] and Omni2[2], whose test sets are derived from the same construction process as their training data, PEBench is independently constructed: the training prompts are generated by comparing pre-edit and post-edit image pairs to describe the difference, whereas the test prompts are generated by GPT-5 from a single source panorama without access to any target image. The test images are also sourced from Internet photos and SkyBox-generated panoramas, disjoint from the training sources.
>
> ## #2. Data composition details:
>
> We provide detailed descriptions of the dataset construction and analysis in Appendix Sec. B. Our data consist of two components:
>
> **(1) PEBench.** Its construction and diversity are analyzed in Appendix Sec. B.1. and B.2.
>
> **(2) Training dataset.** The training data are constructed separately from PEBench, using public panoramic datasets (Structured3D, Pano360, SUN360) together with custom UE-rendered scenes. Each stage uses disjoint public data; details are summarized below:
>
> | Stage | Public | UE | Total | Purpose |
> |---|---|---|---|---|
> | Stage 1 | 12,000 | 2,000 | 14,000 | Text-to-panorama generation |
> | Stage 2 | 12,000 | 2,000 | 14,000 | Controllable generation |
> | Stage 3 | 12,000 | 4,000 | 16,000 → 13,224 | Editing pair synthesis |
>
>
> **(3) Selection and filtering.** For the training data, Stages 1–3 use disjoint, randomly sampled public subsets. In Stage 3, multiple annotators independently review each pair via a spherical viewer from multiple viewpoints, retaining only unanimously approved pairs (13,224 of 16,000). For PEBench, candidate images are randomly sampled and manually screened for quality only (e.g., low resolution), not to exclude difficult cases.
>
>
> ## #3: Method novelty distribution.
>
> We respectfully clarify that Sec. 3.2 is not merely a data processing workflow—it introduces an ERP-native training paradigm with non-trivial design decisions, each validated by ablation.
>
> **Why this is more than a standard data pipeline.** Concurrent methods such as SE360[1] and Omni²[2] construct editing pairs by editing cube faces, which introduces cross-face inconsistencies and limits supported edit types to addition and removal only. Our ERP-native data engine directly synthesizes objects in ERP, supporting all five edit types with consistent supervision—a capability that no existing pipeline provides.
>
> **What Sec. 3.2 contributes beyond data engineering.** Two aspects are methodological: **(1) Geometry-aware generation in ERP (Sec. 3.2.1).** Unlike InstructPix2Pix, which synthesizes pairs in the perspective domain, our controllable generator must produce objects with latitude-dependent ERP distortion. Tab. 3 shows that removing Stage 2 degrades O from 8.01 to 7.50, confirming it provides a geometric prior unavailable from global generation alone. **(2) Layered output design.** Rendering each object on a separate transparent canvas is an architectural decision that enables the Layered Shape Loss (LSL) in Sec. 3.3. Without it, LSL cannot be applied—Sec. 3.2 and 3.3 are tightly coupled.
>
>
> ## #4: Baseline Retraining Details.
>
> All six open-source baselines in Table 1 of the main paper were retrained on our paired editing data (at resolution 512 × 1024) using their official codebases and hyperparameters with LoRA rank 32. The two proprietary methods (GPT-5, Nano Banana Pro) were evaluated via their official APIs with identical prompts. We will include these details in the revision.
>
>
> ## References
>
> [1] SE360: Semantic Edit in 360° Panoramas via Hierarchical Data Construction. AAAI 26.
>
> [2] Omni²: Unifying Omnidirectional Image Generation and Editing in an Omni Model. ACM MM 25.

---

> > ### Author Rebuttal · Reviewer_MTsA · 2026-04-03
> >
> > Thank you for the rebuttal. Most of my concerns have been addressed with detailed explanations and data. The authors claim that Section 3.2 is important and non-trivial, but the method, and the writing organization are not clean enough. It feels as though Section 3.2 merely serves as a supplement to the data construction in Section 3.3, rather than proposing a new algorithm or architecture — it simply feeds additional inputs (e.g., layout) into existing tools (e.g., GPT-5) to meet the task requirements. As a result, its technical contribution to the field remains limited (in other words, Section 3.2 itself has low technical substance, despite complexity and difficulty). That said, the method is useful, and I hope the authors can present it with clearer and more reasonable writing.

---

> > > ### Author Response · Authors · 2026-04-03
> > >
> > > We sincerely thank Reviewer MTsA for confirming that the concerns have been fully resolved, and for the constructive suggestion on improving the presentation of Section 3.2.
> > >
> > > We agree that the current writing could better highlight the technical contributions of Section 3.2, and we will reorganize this section in the revision. We would like to respectfully clarify that Section 3.2 involves non-trivial technical design beyond data processing:
> > >
> > > **[A] ERP-native data synthesis via a custom-trained generator**.
> > > Rather than relying on existing text-to-image models to construct editing pairs—as InstructPix2Pix [Brooks et al., CVPR 2023] does with Stable Diffusion—we train a dedicated controllable generation model directly in the ERP domain. This is necessary for two reasons: (i) purely text-based control inevitably causes unintended background changes (Fig. 3 in their paper); (ii) perspective-domain models are not designed for panoramic images and cannot handle ERP geometric distortion. Our generator produces paired data with precise spatial control, stable backgrounds, and correct latitude-dependent distortion that off-the-shelf models cannot provide.
> > >
> > > **[B] Input-output design of the controllable generator.**
> > >
> > > (1) Input: We adopt full ERP + bounding box + reference image. Unlike concurrent methods (SE360, Omni²) that edit on cube faces, this avoids cross-face seam artifacts and enables flexible spatial/appearance control, supporting all five editing types—including movement and modification that cube-face methods cannot achieve.
> > >
> > > (2) Output: Each object is rendered on a separate transparent layer. This enables the Layered Shape Loss (LSL) in Sec. 3.3 and allows per-object distortion learning—validated by ablation (Tab. 3, O-score: 7.50 → 8.01). Sec. 3.2 and 3.3 are thus co-designed as an integrated framework.
> > >
> > > We will revise the writing of Sec. 3.2 accordingly to better reflect these design contributions.

---

### Official Review · Reviewer_2U8y · 2026-03-13

**Soundness:** 2
**Presentation:** 1
**Significance:** 2
**Originality:** 3
**Overall Recommendation:** 3
**Confidence:** 4

**Summary:**

This manuscript presents a method for panorama generation and editing. Starting from a pre-trained text-to-image model, the authors employ a multi-stage training curriculum to progressively adapt it for panorama editing. They also introduce attention modulation and a shape loss applied to each object layer to improve geometric consistency. The resulting model demonstrates strong editing performance across five tasks on a newly constructed benchmark.

**Compliance With Llm Reviewing Policy:**

Affirmed.

**Final Justification:**

Please see the rebuttal acknowledgement for details.

**Key Questions For Authors:**

See weaknesses. While the paper solves an interesting problem and demonstrates impressive results, the writing lacks rigor and clarity, and the manuscript requires major revision beyond what can be addressed in the rebuttal.

**Limitations:**

The paper includes a discussion on the limitations of the work.

**Strengths And Weaknesses:**

### Strengthes

- The paper solves an interesting problem. The model supports panorama generation given either a text prompt or a perspective image, as well panorama editing across five tasks.
- The model demonstrates strong editing performance.

### Weaknesses

- The paper is very difficult to follow. The presentation is highly disorganized and key motivations and design choices are not clearly explained.
  - **Editing pair construction.** How are the editing pairs constructed? According to Section 3.2, the model first learns to generate panoramas from text-panorama pairs (which could be obtained from panorama datasets and captioned by a VLM). This model is then used as a data engine to construct panorama editing pairs. However, the subsequent controllable generation stage appears to be trained on source–target panorama pairs with layered RGBA object images as intermediate supervision signals. It is unclear where this data comes from and why a synthetic data engine is needed if such rich data already exists, as it could directly support training an editing model.
  - **Bounding box inputs.** Why are bounding boxes included as a model input (Section 3.2.1)? While they might help guide object insertion during the second training stage (controllable generation), they are not used by the final model during panorama generation or editing. This raises questions about whether they provide any value.
  - **Multiple RGBA outputs.** How does the model predict multiple RGBA object layers alongside the full panorama target during the second stage of training? The base model (Qwen-Image) does not natively support this, and the paper does not describe any architectural modifications needed to enable such outputs.
  - The motivation behind attention modulation and layerwise object loss is not well articulated, and the math in Equation 1 and 2 is not well explained.

- Most generation and editing results involve scenes without humans. While this may reflect the bias in the training data, it would be interesting to test the OOD behavior of the model, especially given that the base model was trained on a broad data distribution.

---

> ### Author Rebuttal · Authors · 2026-03-31
>
> We sincerely thank the reviewer for the detailed feedback. We acknowledge the presentation issues and commit to a major revision for clarity. Below we address each technical concern.
>
> ## #1: Writing clarity
>
> We fully accept this criticism. In the revision, we will: (a) restructure Section 3 with a clearer data-flow diagram, (b) add a table summarizing each stage's inputs, outputs, and supervision, and (c) move implementation details to the Appendix to free space for better exposition.
>
> ## #2: Where does the training data come from, and why is a data engine needed?
>
> **(1) Editing-pair construction.** As described in Sec. 3.2.2, our controllable generation model G inserts specified objects into source panoramas at desired locations. We use this capability to construct (source, edit prompt, target) triplets for different operations (e.g., movement pairs are created by placing the same object at different locations), with GPT-5 generating the corresponding edit prompts.
>
> **(2) Training data for the controllable generation model.** The "source–target panorama pairs with layered RGBA object images" mentioned in the review are used to train G, not the final editing model. Details are in Appendix Sec. B.3.3.
>
> **(3) Why is a synthetic data engine necessary?** The RGBA training data for G inherently only cover insertion and removal, since the training objective is to insert an object into a source panorama. Other operations—replacement, movement, modification—require paired images where the same scene differs in object identity, position, or appearance, which do not exist at scale. We therefore use G to synthesize editing pairs for all five edit types.
>
>
>
> ## #3: Why bounding boxes as input?
>
> Bounding boxes serve two purposes: **(1) Paired-data construction.** They provide explicit control over object location and scale, enabling systematic construction of editing pairs for location-sensitive operations (e.g., replacement and movement). **(2) Internalizing spatial understanding.** While boxes are only used as explicit input during Stage 2, the progressive task transition allows the model to internalize this spatial prior. The final editing model learns to infer appropriate object locations from text alone, without requiring box input at inference.
>
>
> ## #4: How does the model predict multiple RGBA layers?
>
> This design is detailed in Appendix Sec. C.2 and illustrated in Fig. 1 of the link(https://anonymous.4open.science/r/World-Shaper-ICML). In Stage 2, the panorama and each object layer are independently VAE-encoded, noised, and patchified, then concatenated into a unified token sequence. The key adaptation is 3D-RoPE with an additional layer-id axis, allowing the transformer to distinguish global panorama tokens from object-layer tokens in a single forward pass.
>
>
> ## #5: Motivation and explanation of attention modulation and layerwise object loss.
>
> **(1) Motivation.** Both components learn distortion-aware object geometry in ERP but at different levels: attention modulation steers cross-attention with GT masks and latitude-dependent strength at the feature level, while layerwise object loss supervises object layers with mask alignment and appearance reconstruction at the output level.
>
> **(2) Explanation of Eq. (1)–(2).** We steer cross-attention A to concentrate inside the object mask M and suppress outside, by interpolating toward A_tar = M ⊙ A_max + (1−M) ⊙ A_min:
>
> A' = (1−α) ⊙ A + α ⊙ A_tar = A + α ⊙ (M ⊙ (A_max − A) − (1−M) ⊙ (A − A_min))
>
> which is Eq. (1). In Eq. (2), α(y) = 1 − cos(φ), where φ is the latitude at row y. At the equator (φ=0), α=0 and attention is unmodified; toward the poles (|φ|→π/2), α→1 and correction is strongest, compensating for ERP stretching at high latitudes.
>
> ## #6: Human-centric OOD evaluation.
>
>
> Following the reviewer's suggestion, we construct *PEBench-Human* to evaluate OOD behavior on human-centric scenes. We collect 500 indoor and 500 outdoor human-containing panoramas from SUN360, with editing targets restricted to humans (e.g., moving a person, changing pose, modifying clothing), yielding 5,000 test cases. Quantitative results are reported below and qualitative examples in Fig. 5 in the link(https://anonymous.4open.science/r/World-Shaper-ICML). Our method generalizes well, outperforming all baselines by a large margin.
>
>
> | Metrics | Qwen | Kontext | GPT-5 | Nano Banana Pro | IC-Edit | Omni² | Step1X-Edit | OmniGen2 | Ours |
> |---|---|---|---|---|---|---|---|---|---|
> | FID ↓ | 93.63 | 90.70 | 123.61 | 75.02 | 116.17 | 87.84 | 108.28 | 95.18 | **50.15** |
> | CLIP_dir ↑ | 9.41 | 8.03 | 7.84 | 11.57 | 10.90 | 11.27 | 7.06 | 8.24 | **18.21** |
> | SC ↑ | 3.44 | 1.39 | 0.41 | 3.38 | 3.25 | 3.36 | 2.99 | 3.49 | **7.62** |
> | PQ ↑ | 2.93 | 1.76 | 2.54 | 2.91 | 2.71 | 2.84 | 2.97 | 2.65 | **7.45** |
> | O ↑ | 3.29 | 0.74 | 0.68 | 3.25 | 3.08 | 3.20 | 3.04 | 3.12 | **7.55** |

---

> > ### Author Rebuttal · Reviewer_2U8y · 2026-04-04
> >
> > Thanks for the rebuttal. It has addressed my main concerns, though ablation experiments on several components would further clarify the value of their design. While I am less worried about the method’s effectiveness (the results are indeed quite impressive), the paper in its current form is very disorganized and does not clearly convey its key contributions or design choices, as other reviewers have noted. I am raising my score to weak reject to acknowledge the effort the authors put into the paper and rebuttal, but I believe the work requires substantial revision and another round of review before it is ready for publication.

---

> > > ### Author Response · Authors · 2026-04-05
> > >
> > > Thank you for the thoughtful follow-up and for acknowledging that our rebuttal addressed your main concerns.
> > >
> > > ## Q1: Further ablation study.
> > > Following your suggestion, we have strengthened the ablation study to better isolate the contribution of each part of the framework.
> > >
> > > First, we ablate the core geometry-aware learning design, including DAAM, LSL, and stage-wise curriculum training (SCT). As shown in Table R2 in the link, each of these components individually brings substantial gains over the Qwen baseline (O: 5.66 → 7.10–7.45), and their combination yields the largest improvement (O: 5.66 → 8.01, +41.5%).
> > > Second, we further ablate the auxiliary refinement modules, including EDM, SDN, BCI, and 3D-RoPE. As shown in Table R1, their combined contribution is relatively modest (O: 8.01 → 8.19, +2.2%). These results clarify that the main performance gains come from the geometry-aware learning design, while the other modules mainly provide complementary refinements.
> > >
> > > ## Q2: Revision plan for improving clarity and paper organization
> > >
> > > We will substantially revise the paper to improve both clarity and organizatio and commit to completing all of these revisions before deadline.
> > >
> > > **Restructuring Plan:**
> > >
> > > *(1) Abstract:*
> > > Being able to edit panoramic images is crucial for creating realistic 360° visual experiences. However, existing perspective-based image editing methods fail to model the spatial structure of panoramas. Conventional cube-map decompositions attempt to overcome this problem but inevitably break global consistency due to their mismatch with spherical geometry. Motivated by this insight, we reformulate panoramic editing directly in the equirectangular projection (ERP) domain and present World-Shaper, **a unified geometry-aware framework that supports five distinct editing operations within a single ERP-native representation. To address the latitude-dependent geometric distortion inherent in ERP, we introduce a geometry-aware learning strategy comprising distortion-aware attention modulation (DAAM), which steers cross-attention with latitude-dependent strength at the feature level; layered shape loss (LSL), which enforces per-object geometric supervision at the output level; and progressive curriculum training to internalize panoramic priors. To overcome the scarcity of paired panoramic editing data, we train a dedicated ERP-native controllable generator that synthesizes objects directly in the equirectangular domain conditioned on bounding boxes and reference images, with each object rendered on a separate transparent layer to enable both diverse paired data construction and layered geometric supervision.** Extensive experiments on our new benchmark, PEBench, demonstrate that World-Shaper achieves superior geometric consistency, editing fidelity, and text controllability compared to state-of-the-art methods, enabling coherent and flexible 360° visual world creation with unified editing control.
> > >
> > > *(2) Contribution List:*
> > > 1. We introduce World-Shaper, a unified framework that supports five distinct panoramic editing operations within a single ERP-native representation, ensuring global consistency and seamless cross-view editing.
> > > 2. We propose a geometry-aware learning strategy to handle latitude-dependent distortion in ERP, comprising (i) distortion-aware attention modulation (DAAM), which steers cross-attention with latitude-dependent strength at the feature level; (ii) layered shape loss (LSL), which enforces per-object geometric supervision at the output level; and (iii) progressive curriculum training to internalize panoramic priors.
> > > 3. We curate PEBench, a comprehensive panoramic editing benchmark, and conduct extensive experiments to demonstrate the superiority of our method.
> > >
> > > *(3) Paper Structure:*
> > > - **Introduction** — Foreground geometry-aware learning as the central contribution.
> > > - **Related Works** — Add positioning with InstructPix2Pix and ControlNet-style conditioning; sharpen positioning against CubeDiff and HunyuanWorld.
> > > - **Method**
> > >   - **3.1 Overview**
> > >   - **3.2 Geometry-Aware Learning Strategy** — Promoted from original Sec. 3.3 as the central technical section.
> > >     - **3.2.1 DAAM** — expand motivation and derivation.
> > >     - **3.2.2 LSL** — add RGBA output details and motivation.
> > >     - **3.2.3 Progressive Curriculum Training**
> > >   - **3.3 ERP-Native Data Synthesis** — Demoted from original Sec. 3.2 as supporting infrastructure.
> > >     - **3.3.1 Controllable Panorama Generator** — Clarify input/output design and the dual role of bounding boxes.
> > >     - **3.3.2 Paired Data Construction** — Cover all five edit types and add data composition details.
> > > - **Experiments**
> > >   - **4.1 Evaluation Protocol** — Add FID_ROI, CLIP_ROI, and FAED; expand PEBench and use alternative LLMs to eliminate circularity.
> > >   - **4.2 Comparisons with SOTAs** — Adopt enhanced prompts for all baselines.
> > >   - **4.3 Ablation Study** — Add core component ablations and stage ablations (S1/S2/S3).
> > > - **Conclusion**

---

### Decision · Program_Chairs · 2026-04-30

**Decision:**

Accept (regular)

**Comment:**

The submission received three Weak Accepts​ and one Weak Reject​ as final recommendations.

Reviewers appreciated the interesting geometry-aware framework​ and acknowledged that PEBench is a valuable contribution. Concerns were raised regarding clarity of presentation, heavy GPT-5 dependency, and the framework’s large number of components.
Following a strong rebuttal, most concerns have been addressed. The issue of presentation clarity, however, remains.
Given that the submission demonstrates a clear contribution and the presentation can be further improved during revision, AC acceptance at this stage.

The authors are required to integrate all new experiments and clarifications from the rebuttal into the final manuscript. Please ensure the material is clearly organized and presented in the camera-ready version.